# Circuit-Level Steering for Personalized Knowledge Injection in Language Models

## Abstract

As large language models (LLMs) become central to user-facing applications, effective personalization, adapting models to individual users' evolving facts and contexts, has become crucial. However, existing approaches struggle with **mutable personal knowledge**: finetuning can embed static user information but is costly and prone to catastrophic forgetting, while knowledge editing methods rely on pre-cached representations from large corpora like Wikipedia, which are unavailable or unsuitable for personal domains due to data scarcity and privacy concerns. We formalize updating the fact-level personalization with mutable knowledge as a new task, constructing synthetic Personal Knowledge Graphs (PKGs) that capture user information across time points to evaluate models' ability to incorporate updates without degrading existing knowledge. Drawing on insights from mechanistic interpretability, we discover that personal facts are encoded in localized circuits within LLMs. We propose **SPIKE** (Steering for Personalized Knowledge Injection), which combines adapter modules with steering-based activation injection, targeting identified personal knowledge circuits. This approach enables the precise integration of new user-specific facts, including previously unseen triples, while maintaining the integrity of prior knowledge. Our experiments demonstrate that SPIKE effectively balances the accuracy of incorporating new facts with the preservation of existing knowledge, offering a practical solution for continual personalization in settings where user information evolves frequently.

## 1 Introduction

As large language models (LLMs) become increasingly integrated into user-facing applications, **personalization**, which adapts the model to reflect the facts, contexts, and preferences of individual users, has emerged as a critical direction for practical AI. Current LLM personalization techniques mainly focus on aspects such as persona modeling or writing style adaptation (Jiang et al., 2024; Liu et al., 2025; Li et al., 2025; Zhang et al., 2025). However, these methods may fall short when it comes to reasoning over *grounded, user-specific **factual** knowledge*, such as "Mike started commuting by bike instead of taking the subway." or "Jack transitioned from being a programmer to working as a product manager.". Addressing personalized factual knowledge is essential to advance LLMs toward the role of personal agents. For example, an LLM capable of accurately answering (reasoning) a user's waking preferences could autonomously set individualized alarms. Consequently, it becomes necessary to explore approaches that **internalize** such knowledge within the LLM itself, enabling reasoning that is both accurate and contextually aligned with the user.

Personal factual knowledge, however, poses unique challenges: it is inherently **mutable**, reflecting changes such as job transitions, address updates, or evolving daily routines, and also comes with natural privacy concerns. One approach to integrate *mutable personal factual knowledge* into LLMs is to *finetune* them on personal data (Dutt et al., 2022; Salemi et al., 2024), embedding user-specific information directly into model parameters. While this can embed personal knowledge, the approach is computationally costly and risks **catastrophic forgetting**, potentially degrading the model's global capabilities (Dou et al., 2024). Another line of research, *knowledge editing* methods such as ROME (Meng et al., 2022) or MEMIT (Meng et al., 2023), updates facts in the model by leveraging pre-cached representations obtained from large-scale corpora like Wikipedia. However, this assumption does not hold in the personal domain: for each individual, there is no large corpus

from which to derive such pre-cached structures, and even if available, repeated extraction of sensitive user data would pose serious privacy concerns. These limitations highlight the need for new strategies to effectively internalize mutable personal factual knowledge.

In this work, we formalize *fact-level personalization with mutable knowledge* as a new LLM task, aiming to effectively internalize mutable personal knowledge in LLMs, updating changed facts while preserving unchanged personal knowledge.

We first assume that the original personal information to be internalized is stored as a Personalized Knowledge Graph (PKG). This aligns with recent work using KGs as external memory due to their **modular, interpretable, and easily updatable** nature (Dutt et al., 2022; Wang et al., 2024b; Prahlad et al., 2025). However, rather than relying on external memory, we aim to *internalize* the KG into the LLM model. To effectively internalize personal information into LLMs, two key objectives must be clearly defined: (i) where within the model's parameter space the modifications should occur, and (ii) how those parameters should be updated.

For the first objective, our insight is based on the concept of **knowledge circuits** (Yao et al., 2024), which posits that different domains of knowledge are handled by different components in an LLM. We show that *personal facts are encoded in localized circuits*, and propose a **circuit-aware injection strategy** that targets only the relevant substructures. Empirical analysis on personalized questions indicates that adjusting the parameters of attention heads provides a more effective means of updating knowledge than modifying the FFN (Feed Forward Neural Networks) parameters.

For the second objective, we construct a representation that captures the discrepancy between the updated information and its initial information, which is incorporated through the identified personal knowledge circuits. Unlike existing editing methods that rely on pre-cached representations from large corpora, our approach derives this signal directly from the structured personal knowledge itself. This design minimizes unnecessary parameter changes and mitigates unintended side effects, while enabling precise integration of user-specific information. This method, which we refer to as circuit-level **S**teering for **P**ersonal**I**zed **K**nowledge inj**E**ction in language models (**SPIKE**), enables precise and efficient integration of user-specific updates directly into the relevant model circuits. Our method strikes a balance between **accurately integrating new facts** (accuracy) and **preserving the integrity of prior knowledge** (locality). Furthermore, our method extends beyond conventional knowledge editing settings by enabling LLMs to incorporate *unseen updated triples* without disrupting existing knowledge. Our main contributions are summarized as follows:

- We introduce a new task setting for LLM personalization that models **updates of user-specific knowledge** over time in the form of changing knowledge graph triples.

- We leverage insights from mechanistic interpretability to identify and target the **responsible circuits for personal knowledge**, enabling effective and localized editing within the LLM.

- We propose a method that allows the LLM to integrate updated triples, **reflecting new user-specific facts without compromising prior knowledge**.

## 2 RELATED WORK

### 2.1 KNOWLEDGE EDITING METHODS

Knowledge editing methods modify LLM parameters without full re-training, but this risks side effects such as forgetting or distortion (Gupta et al., 2024). Many approaches also assume access to pre-cached representations from large corpora (e.g., Wikitext), an assumption that fails in the personal domain due to data scarcity and privacy concerns. Representative methods include ROME (Meng et al., 2022), which edits FFN parameters as key–value memories, MEMIT-Merge (Dong et al., 2025) extending MEMIT (Meng et al., 2023) to support batch edits for the same subject, and AlphaEdit (Fang et al., 2025), which preserves unrelated knowledge through a projection step. These illustrate the potential of editing but also its limitations for adapting LLMs to evolving personalized KGs.

## 2.2 CIRCUIT FINDING

Recent efforts to interpret Transformer-based large language models (LLMs) have focused on identifying compact subgraphs of the model that are responsible for specific behaviors. These subgraphs, known as circuits, typically consist of a small set of attention heads and MLPs that strongly influence the output. Automated Circuit Discovery (ACDC) (Conmy et al., 2023) formalizes circuit extraction as a subgraph selection problem, where nodes are LLM components and edges denote their connections. By progressively removing low-impact edges, it produces compact, interpretable circuits, but at a high computational cost since each edge requires a separate forward pass. Edge Attribution Patching (EAP) (Syed et al., 2024) mitigates this with a gradient-based approximation. HeadMap (Wang et al., 2025) instead ranks attention heads by their contribution, retaining only the most influential ones for fine-tuning. This reduces overhead, avoids unnecessary updates, and maintains interpretability, making it a practical alternative to full-model tuning.

## 3 PRELIMINARIES

### 3.1 TASK FORMULATION

Personal factual knowledge is not static: individuals frequently change their occupations, addresses, or daily routines. A practical system must not only reflect newly updated information but also preserve consistency with personal knowledge that remain unchanged. Motivated by this scenario, we formulate a fact-level personalization task that explicitly models updates in personal knowledge graphs (PKGs). We assume access to two versions of a PKG, which serves as a minimal and structured interface to personal information: the initial KG ($\mathbf{KG}^{\text{init}}$) and the updated KG ($\mathbf{KG}^{\text{upd}}$). The initial KG contains user-specific facts that have already been internalized into the LLM via fine-tuning. The updated KG reflects changes that occur over time (modeled here as a single update for simplicity), such as a new workplace or altered daily routine (See Appendix A.2 for the formal formulation).

A practical example of our task is as follows: When the KG itself contains sensitive individual-level information (e.g., medical histories or financial transactions), external retrieval from the KG poses direct privacy risks. Even partial disclosure may expose identifiable attributes of individuals (e.g., Patient C) and breach confidentiality. By internalizing the personalized KG into the LLM through fine-tuning or adaptation, the model can answer personal queries without exposing raw records at *inference time*. This makes internalization an effective mechanism for safeguarding privacy while enabling personalized responses.

### 3.2 PERSONAL KNOWLEDGE GRAPH CONSTRUCTION

To experiment with the task formulation introduced earlier, we construct two personalized KG datasets, **PeaCoK-Ex** (Extended) and **PerInfoKG**.

**PeaCoK-Ex** extends the original PeaCoK (Gao et al., 2023) (based on commonsense knowledge), adding relations for personal attributes (e.g., `experience`, `routine_habit`, `characteristics`) and introducing 822 synthetic individuals, each linked to a single occupation. The resulting KG contains 105K triples, 49K entities, and 18 relations. To build $\mathbf{KG}^{\text{upd}}$, we modify 20 % of the person–occupation pairs while keeping other attributes fixed.

**PerInfoKG** is a dataset we create by defining 23 personal information fields for each of 2,000 fictitious individuals, resulting in 46K triples and 2,134 entities in total. For every individual, we partition the 23 fields into 17 *mutable* attributes used for editing and 6 *immutable* attributes reserved for evaluating locality, so that each individual contributes to both edit and locality evaluation. We prepare two versions of this dataset: (i) an **edit setting** where 200 individuals are randomly sampled and the required update triples are directly provided as supervision for injection (editing), and (ii) an **unseen test setting** (Section 5.4.1) where all 2,000 individuals are used to train the alignment module, and generalization is evaluated on previously unseen cases.

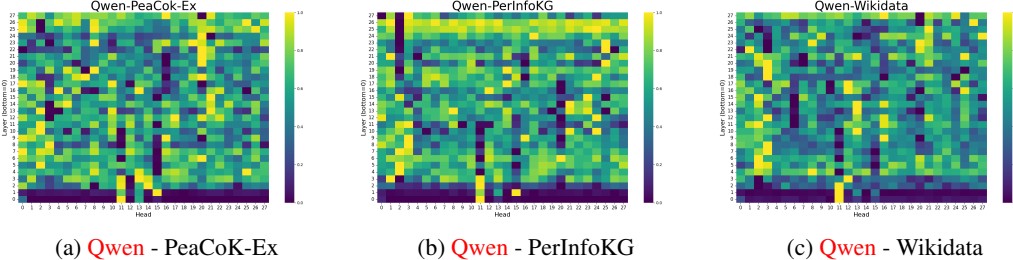

(a) Qwen - PeaCoK-Ex       (b) Qwen - PerInfoKG       (c) Qwen - Wikidata

Figure 1: Heatmap visualizations of important components identified in Qwen2.5-7B-Instruct (Yang et al., 2024) across PeaCoK-Ex, PerInfoKG, and Wikidata (Meng et al., 2022) (i.e., *Known1000* dataset). Each plot shows attention heads ($x$-axis) by layers ($y$-axis). The similarity is highest when comparing personal knowledge datasets.

## 4 METHODOLOGY

### 4.1 CIRCUIT ANALYSIS FOR PERSONAL INFORMATION LOCALIZATION

To efficiently reflect updated personal information in large language models (LLMs) while minimizing side effects such as unintended changes to unrelated knowledge, we adopt a circuit-level analysis approach inspired by mechanistic interpretability. This perspective enables us to identify and intervene on a sparse subset of components that are causally linked to personal knowledge, thereby avoiding the need for full-model fine-tuning.

We first investigate which components of LLMs, attention heads or FFNs, are more effective targets for updating personal knowledge. To this end, we extend HeadMap (Wang et al., 2025), which quantifies the importance of attention heads by masking their outputs and measuring the induced loss, to FFNs using the same strategy. Based on these scores, we select a limited number of parameters (heads or FFNs across layers) for selective fine-tuning with personal knowledge from PeaCoK-Ex. On Qwen2.5-7B-Instruct (Yang et al., 2024), fine-tuning only 3 important layers

Table 1: Selective Finetuning Results on Qwen2.5-7B-Instruct

| Target | Ratio (%) | Acc (%) |
|--------|-----------|---------|
| FFN | 2.67 | 36.64 |
| Heads | 2.70 | 98.95 |

of FFNs (2.67% of total parameters) yields just 36.64% accuracy, while finetuning a similar number of parameters corresponding to important attention heads, which involves 3 heads per layer across all layers (2.7% of total parameters), the model achieves 98.95% accuracy (See Table 1). These results demonstrate a clear distinction: FFNs appear to require broad, costly intervention to be effective, whereas attention heads enable efficient and accurate updates when targeted selectively in personalized factual queries.

Based on this finding, we focus our circuit discovery efforts solely on attention heads. We identify circuits responsible for personalization based on importance scores of attention heads and present a heatmap visualization of layer-wise attention head importance scores for Qwen2.5-7B-Instruct in Figure 1. For Qwen2.5-7B-Instruct, the similarity between the personal information datasets PerInfoKG and PeaCoK-Ex is 0.6242, which is higher than the similarity between PerInfoKG and Wiki-based general knowledge (Wikidata, 0.5335), suggesting the presence of circuits dedicated to personal information. Detailed procedures for computing similarity, as well as performance tables for additional models, are provided in Appendices A.5 and A.6. Although personalized circuits may appear more appropriate than a single global circuit, they introduce scalability challenges because a new circuit must be computed for every incoming user. Additional analysis and a detailed comparison between personal and global circuits are provided in the Appendix A.6

### 4.2 PERSONAL INFORMATION INJECTION MODULE

Building on the circuit identified in Section 4.1, we propose a steering mechanism (Rimsky et al., 2024) that enables an LLM to reflect updates from $\textbf{KG}^{upd}$ while preserving previously encoded knowledge. The key idea is to align structured user-profile facts (represented as triples) with internal LLM representations and steer the model by intervening on a sparse set of attention heads (we select

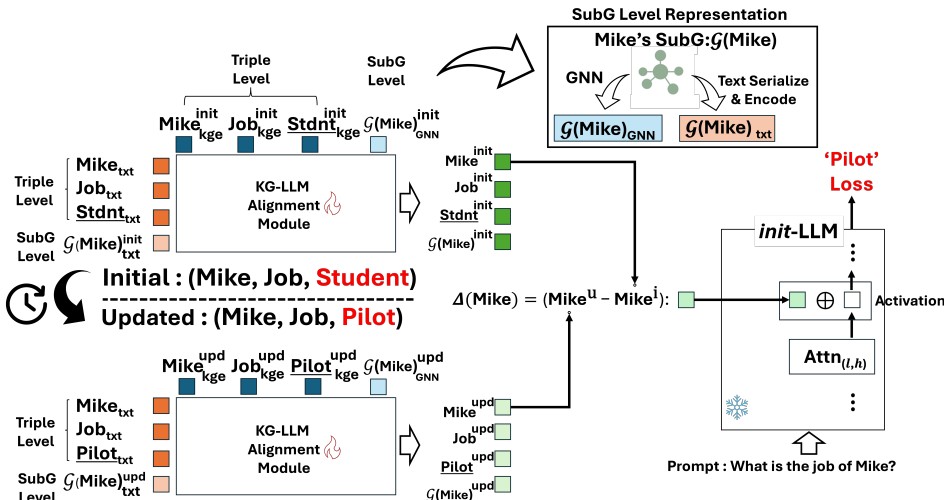

Figure 2: Illustration of our knowledge injection process when (Mike, Job, Student) triple is updated to (Mike, Job, Pilot).

the top-$k$ most important heads based on importance scores for each layer), rather than modifying all parameters.

Figure 2 provides an overview of this pipeline and visually illustrates the update process. The following update scenario can illustrate the overall process. From the initial KG, the occupation of a person Mike is Student, which is later updated to Pilot. To incorporate this change, we analyze the difference between the two triples ((Mike, Job, Student) vs. (Mike, Job, Pilot)), obtain their textual representations through the alignment procedure (Section 4.2.1), and use the resulting difference to steer the outputs of the attention heads identified by our circuit analysis (Section 4.1). This circuit-guided intervention allows the LLM to behave as if it had internalized the updated information (Section 4.2.2). Since the steering signal is derived from the difference between $\textbf{KG}^{\text{init}}$ and $\textbf{KG}^{\text{upd}}$, knowledge that remains unchanged exerts little or no influence on the model's behavior.

### 4.2.1 KG-LLM ALIGNMENT MODULE

When updates occur in the profile, the KG-LLM Alignment Module aligns the two representation spaces so that the LLM can generate responses consistent with the modified information.

The alignment is performed at two levels. The first is **triple-level alignment**, which focuses on local updates of individual triples (e.g., (Mike, Job, Student) → (Mike, Job, Pilot)). The second is **subgraph-level alignment**, which captures broader structural changes within the local subgraph centered on the updated entity $\mathcal{G}(\text{Mike})$. The motivation for separating the two levels is that triple-level updates alone cannot capture higher-order differences that arise in the overall subgraph structure (Wang et al., 2020).

The inputs to the alignment module consist of two modalities: structural representations from the KG and textual representations from the LLM. On the KG side, we include both triple-level embeddings extracted from a pretrained KG embedding model Bordes et al. (2013); Trouillon et al. (2016) and subgraph-level embeddings obtained from a GNN encoder (Wang et al., 2020) over the local subgraph centered on the updated entity. On the LLM side, we obtain textual representations by serializing the triple or subgraph into natural language and encoding them with the LLM.

The alignment module follows an attention-based structure, where textual representations serve as queries, structural embeddings as keys, and values:

$$\mathbf{Q_{txt}}, \mathbf{K_{kge}}, \mathbf{V_{kge}} = \mathbf{H_{txt}}\mathbf{W_Q}, \mathbf{H_{kge}}\mathbf{W_K}, \mathbf{H_{kge}}\mathbf{W_V}, \tag{1}$$

where $\mathbf{W_Q}, \mathbf{W_K}, \mathbf{W_V} \in \mathbb{R}^{D \times d}$. Here, $\mathbf{H_{txt}}$ and $\mathbf{H_{kge}}$ are defined as:

$$\mathbf{H_{txt}} = [\mathbf{h}_{txt}, \mathbf{r}_{txt}, \mathbf{t}_{txt}, \mathbf{h}_{txt}^{\mathcal{G}(h)}] \in \mathbb{R}^{4 \times D}, \ \ \mathbf{H_{kge}} = [\mathbf{h}_{kge}, \mathbf{r}_{kge}, \mathbf{t}_{kge}, \mathbf{h}_{kge}^{\mathcal{G}(h)}] \in \mathbb{R}^{4 \times D}. \quad (2)$$

Here, $\mathbf{h}_{txt}, \mathbf{r}_{txt}, \mathbf{t}_{txt}$ correspond to the textual embeddings of the head(h), relation(r), and tail(t) of given knowledge graph, while $\mathbf{h}_{txt}^{\mathcal{G}(h)}$ denotes the subgraph textual embedding of head(h). Similarly, $\mathbf{h}_{kge}, \mathbf{r}_{kge}, \mathbf{t}_{kge}$ correspond to the KG embeddings of the triple, and $\mathbf{h}_{kg}^{\mathcal{G}(h)}$ denotes the GNN-encoded subgraph embedding.

The alignment process is then performed as:

$$\hat{\mathbf{H}}'_{\mathbf{txt}} = \mathrm{Softmax}(\mathbf{Q_{txt}}\mathbf{K_{kge}}^T / \sqrt{d})\mathbf{V_{kge}}, \quad \hat{\mathbf{H}}_{\mathbf{txt}} = MLP_{up}(\hat{\mathbf{H}}'_{\mathbf{txt}}), \quad (3)$$

where $d$ denotes the dimensionality of the queries, and $MLP_{up}$ denotes a linear transformation. For both initial and updated KG settings, we obtain aligned textual representations $\hat{\mathbf{H}}_{\mathbf{txt}}^{\mathrm{init}}, \hat{\mathbf{H}}_{\mathbf{txt}}^{\mathrm{upd}} \in \mathbb{R}^{(4 \times D)}$. The first embedding vector of each corresponds to the aligned representation of the updated entity in the textual space (i.e., $\hat{\mathbf{H}}_{\mathbf{txt}}^{\mathrm{init}}[0, :]$ and $\hat{\mathbf{H}}_{\mathbf{txt}}^{\mathrm{upd}}[0, :]$, respectively). These correspond to $\mathbf{Mike}^{\mathrm{init}}$ and $\mathbf{Mike}^{\mathrm{upd}}$ in Figure 2, respectively.

### 4.2.2 Updated Knowledge Adaptation via LLM Steering

The representation difference is computed as the change in the head entity's embedding (position [0]) extracted from the alignment module, and this difference is used to steer the LLM's behavior. Specifically, we compute the difference vector between the two time points, $\Delta = \sigma(\hat{\mathbf{H}}_{\mathbf{txt}}^{\mathrm{upd}}[0, :] - \hat{\mathbf{H}}_{\mathbf{txt}}^{\mathrm{init}}[0, :])$, and inject it into the output of the attention heads identified during circuit discovery. Here, $\sigma$ denotes the sigmoid function. Importantly, knowledge injection is applied only at the heads previously identified as responsible for personal information processing (Section 4.1).

Let $d_{\mathrm{head}}$ denote the dimensionality of a single head output and $N$ is the number of selected heads. The output dimensionality of the alignment module is set to $d_{\mathrm{head}} \times N$, which is partitioned into $N$ segments, each added to the corresponding head output. This design enables us to selectively control the internal activations of the LLM, allowing a model to respond to the updated information.

As a result, our method achieves personalized knowledge updates and generation without requiring full fine-tuning, relying instead on activation steering. In addition, since the alignment module is trained to convert updated triples into LLM representations and inject them into intermediate activations to steer the model's behavior, our approach naturally extends to unseen settings, where updates for previously unseen triples (with seen entities and relations) can still be incorporated effectively (Figure 3a).

### 4.2.3 Optimization

The alignment module is trained so that the injected difference vector accurately transforms the activation from the initial timestamp into a representation consistent with the updated knowledge. Given a pair of initial triple $((s_i, r_i, o_i^{\mathrm{init}}))$ and updated triple $((s_i, r_i, o_i^{\mathrm{upd}}))$, the negative log-likelihood loss is defined as $\mathcal{L}_{\mathrm{NLL}} = -\sum_i \sum_{t=1}^{|o_i^{\mathrm{upd}}|} \log \ p\left(o_{i,t}^{\mathrm{upd}} \mid o_{i,<t}^{\mathrm{upd}}, g_{+\Delta}^{\mathrm{init}}(s_i, r_i)\right)$, where $g_{+\Delta}^{\mathrm{init}}$ denotes the *init-LLM* steered by the difference vector $\Delta = \sigma\left(f_\phi((s_i, r_i, o_i^{\mathrm{upd}})) - f_\phi((s_i, r_i, o_i^{\mathrm{init}}))\right)$, and $f_\phi$ is the alignment module parametrized by $\phi$.

To ensure that knowledge unrelated to the updates remains intact, we introduce a KL divergence loss between the output distributions of the LLM before and after steering, obtained by feeding the subject into the model: $\mathcal{L}_{\mathrm{KL}} = D_{\mathrm{KL}}\left(\mathrm{softmax}\left(g^{\mathrm{init}}(s_i)\right) \ \| \ \mathrm{softmax}\left(g_{+\Delta}^{\mathrm{init}}(s_i)\right)\right)$. Furthermore, to prevent the steering vector ($\Delta$) from deviating excessively from the initial representation $f_\phi((s_i, r_i, o_i^{\mathrm{init}}))$, we add a norm-based penalty: $\mathcal{L}_{\mathrm{norm}} = \frac{\|\Delta\|_2}{\|f_\phi((s_i, r_i, o_i^{\mathrm{init}}))\|_2}$. Finally, the overall training objective is

$$\mathcal{L} = \mathcal{L}_{\mathrm{NLL}} + \lambda_1 \cdot \mathcal{L}_{\mathrm{KL}} + \lambda_2 \cdot \mathcal{L}_{\mathrm{norm}}, \quad (4)$$

where $\lambda_1$ and $\lambda_2$ are scaling factors. The learnable parameters include those of the KG–LLM alignment module ($\phi$) and the GNN encoder responsible for subgraph-level representations.

# 5 EXPERIMENT

## 5.1 EXPERIMENTAL SETUP

Our problem setting (Section 3.1) considers the scenario in which an LLM, initially fine-tuned on the initial personal knowledge graph $\mathbf{KG}^{\text{init}}$, is subsequently updated with the modified knowledge contained in $\mathbf{KG}^{\text{upd}}$. To simulate an LLM that already encodes prior personal information, we first inject $\mathbf{KG}^{\text{init}}$ through supervised fine-tuning, resulting in what we refer to as the *init-LLM*. Since all personal information constructed in Section 3.2 is represented as triples (e.g., (Mike, Medical_Condition, Hypertension)), we design prompt templates for each relation type (e.g., Medical_Condition: "{Subject} suffers from") and fine-tune the model to predict the correct tail entity given the head and relation. To ensure reliability, the resulting *init-LLM* is trained until it achieves over 99% accuracy on $\mathbf{KG}^{\text{init}}$ triples, guaranteeing that the initial personal knowledge is fully encoded before conducting update experiments with $\mathbf{KG}^{\text{upd}}$.

Although one could assume a separate personalized LLM per individual, this is computationally prohibitive as it would require storing a distinct model for every user. Instead, our *init-LLM* is trained to encode the collective personal knowledge of all users. During evaluation, when updating to $\mathbf{KG}^{\text{upd}}$, we inject the changes corresponding to a specific individual, measure performance (Table 2), and then reload the original *init-LLM* before repeating the process for another individual.

It should be noted that in our experimental setup, updates from $\mathbf{KG}^{\text{upd}}$ are applied exclusively through the modified triple set; triples that remain unchanged relative to $\mathbf{KG}^{\text{init}}$ are not re-injected.

### 5.1.1 DATASETS

We evaluate our approach on two personalized knowledge graph (KG) datasets, PeaCoK-Ex and PerInfoKG, constructed in Section 3.2. Each dataset consists of two temporal snapshots: an initial KG (i.e., $\mathbf{KG}^{\text{init}}$) containing both unchanged and updated personal facts, and an updated KG (i.e., $\mathbf{KG}^{\text{upd}}$) reflecting changes to a subset of those facts. The update set corresponds to triples that differ between $\mathbf{KG}^{\text{init}}$ and $\mathbf{KG}^{\text{upd}}$, while the remaining triples stay unchanged and serve as the basis for evaluating *locality*. We inject only the modified triples when updating from $\mathbf{KG}^{\text{init}}$ to $\mathbf{KG}^{\text{upd}}$.

In the PeaCoK-Ex dataset, the only updated personal field is Job, the tail entity of a triple where the relation is has_a_job_of. Once a person's job changes, there are no other unchanged attributes left for that individual, so locality cannot be directly assessed on the same person. Instead, we evaluate locality using the information of other individuals whose facts remain unchanged. In contrast, the PerInfoKG dataset contains 23 personal fields. Thus, even if one field, such as job information, is updated, many other fields for the same person remain intact, allowing locality to be measured directly on that individual by relying on the unaffected fields. Detailed dataset statistics are provided in the Appendix A.3.

### 5.1.2 BASELINES

We compare our approach against several representative baselines for predicting the correct tail entity in $\mathbf{KG}^{\text{upd}}$. These include: (i) full-model fine-tuning (FT), which updates all parameters of the LLM; (ii) circuit-selective fine-tuning (FT-C), which only fine-tunes the personal-knowledge circuit identified in Section 4.1; and (iii) knowledge editing methods that modify parameters in specific layers to update factual knowledge. Among editing methods, we consider four representative approaches. ROME (Meng et al., 2022) treats FFN modules as key–value memories and directly alters them to inject new facts. MEMIT-Merge (Dong et al., 2025) extends MEMIT by merging edits for overlapping subjects, making it effective in batch editing scenarios. Finally, AlphaEdit (Fang et al., 2025) projects updates to avoid interference with preserved knowledge, explicitly supporting *locality*. WISE (Wang et al., 2024a) performs continual editing by storing new knowledge in a dedicated side memory and routing queries accordingly. ICE (Qi et al., 2025) applies consistency-based supervision to align the model's predictions with contextual prompts without relying on hard one-hot targets.

Table 2: Performance comparison of injecting updated personal information into LLMs. Acc., Loc., and Ret. denote Accuracy, Locality, and Retention, respectively. Total Score denotes the harmonic mean of Accuracy, Locality, and Retention.

| Method | LLM Model | PeaCoK-Ex | | | | PerInfoKG | | | |
|---|---|---|---|---|---|---|---|---|---|
| | | Acc. (%) | Loc. (%) | Ret. (%) | Total | Acc. (%) | Loc. (%) | Ret. (%) | Total |
| FT | | **100.00** | 50.55 | 75.09 | 69.61 | **100.00** | 84.27 | 78.31 | 86.62 |
| FT-Circuit | | **100.00** | 46.04 | 74.55 | 66.47 | **100.00** | 70.73 | 89.13 | 84.84 |
| ICE | | **100.00** | 85.59 | **88.23** | 90.86 | 41.01 | 14.56 | 38.67 | 25.23 |
| LoRA | GPT-J (6B) | 89.02 | 65.63 | 83.53 | 78.04 | 99.92 | 63.69 | 54.46 | 68.07 |
| ROME | | 93.90 | 88.88 | 85.88 | 89.43 | 61.96 | 70.05 | **90.35** | 72.32 |
| MEMIT-Merge | | 71.90 | 88.89 | 55.11 | 69.28 | 47.82 | 71.80 | 71.73 | 61.50 |
| AlphaEdit | | 98.78 | 99.83 | 75.21 | 89.72 | 26.10 | 41.96 | 36.25 | 33.43 |
| WISE | | **100.00** | 91.74 | 87.50 | 92.80 | 99.44 | 94.93 | 55.60 | 77.77 |
| Ours | | **100.00** | **99.98** | 86.50 | **95.05** | 99.23 | **99.83** | 84.48 | **93.95** |
| FT | | **100.00** | 34.13 | 93.06 | 59.95 | **100.00** | 67.74 | 93.61 | 84.64 |
| FT-Circuit | | 95.73 | 53.82 | 80.09 | 72.27 | 90.08 | 95.18 | 91.04 | 92.05 |
| ICE | | 99.39 | 88.29 | 91.82 | 92.94 | 73.66 | 42.60 | 42.57 | 49.55 |
| LoRA | Qwen2.5-7B | 91.00 | 68.04 | 86.71 | 80.60 | 93.76 | 67.97 | 73.33 | 76.89 |
| ROME | | 81.71 | 86.63 | **95.10** | 87.47 | 50.73 | 51.81 | **95.13** | 60.57 |
| MEMIT-Merge | | 67.88 | 80.96 | 65.76 | 70.94 | 65.96 | 74.12 | 40.28 | 56.10 |
| AlphaEdit | | 98.78 | **100.00** | 85.60 | **94.32** | 18.42 | 77.62 | 90.34 | 38.34 |
| WISE | | 85.97 | 96.24 | 91.84 | 91.15 | 94.11 | 91.67 | 55.30 | 75.72 |
| Ours | | **100.00** | 92.70 | 89.24 | 93.77 | 99.66 | **96.45** | 84.10 | **92.90** |

## 5.2 EVALUATION METRICS

We evaluate the performance of our method and baselines using three quantitative metrics: Accuracy, Locality, and Retention rate. First, **Accuracy** quantifies the ratio of correctly produced outputs following the injection of modified personal knowledge into the LLM. Second, **Locality** assesses the model's capability to preserve pre-existing, *unchanged personal information* following the update. Finally, **Retention** evaluates the retention of *general world knowledge* that was originally encoded in the *init-LLM*, ensuring that such knowledge is not compromised by the injection of personal information. To construct the evaluation set for Retention, we utilize the known_1000 dataset introduced by ROME (Meng et al., 2022). Specifically, we query the *init-LLM* on this dataset and sample 200 facts that are correctly predicted by *init-LLM*, ensuring that we evaluate the retention of knowledge the model actually possessed prior to editing.

## 5.3 MAIN RESULTS

The performance on the two datasets, PeaCoK-Ex and PerInfoKG, is presented in Table 2. Each dataset exhibits distinct characteristics: the number of updated triples per subject is fixed to 1 in PeaCoK-Ex, whereas it is 17 in PerInfoKG. Consequently, as shown in the table, most baselines achieve relatively high accuracy on PeaCoK-Ex. In contrast, certain approaches, such as finetuning and LoRa, demonstrate weaker performance in terms of Locality. Overall, the ability to retain general knowledge (Retention) largely correlates with Locality, suggesting that preserving unchanged personal information is similar to maintaining general knowledge. From the results, Retention is model-dependent; Qwen generally outperforms GPT-J, reflecting the superior intrinsic capabilities of the backbone LLM. When comparing with datasets, Retention score on PeaCoK-Ex is higher than that on Locality, because sparse single-fact updates exert minimal influence on the broader

model. Conversely, on PerInfoKG, Retention score exhibits a notable decline, dropping to a level similar to Locality on average. This suggests that the dense updates (17 facts per subject) in PerInfoKG significantly perturb model parameters, leading to degradation in both personal locality and general knowledge retention. Furthermore, editing-based approaches generally perform well on the PeaCoK-Ex dataset, since it still contains a large amount of commonsense knowledge and thus remains aligned with the pre-cached representations obtained from large-scale corpora like Wikipedia.

Our experiments reveal a more specific limitation of existing editing models when compared in multiple update scenarios (e.g., PerInfoKG dataset). Most knowledge editing baselines (ROME, AlphaEdit) fail to achieve good performance on both performance measures. One possible candidate reason for this observation is that many editing methods assume access to pre-cached representations derived from large general-domain corpora (e.g., Wikipedia) to guide and stabilize edits. Such representations are unavailable in the personal domain due to both data scarcity and privacy considerations, making these approaches ill-suited for handling mutable personal knowledge.

Another reason relates to structural limitations in handling multiple correlated updates. While several methods allow batch editing across different facts, they do not natively support simultaneous updates to multiple facts tied to the same subject (with the exception of MEMIT-Merge (Dong et al., 2025), which explicitly merges edits for the same subject). For example, when both $(s_1, r_1, o_1)$ and $(s_1, r_2, o_2)$ must be modified together, these models typically treat each edit independently and fail to maintain consistency across correlated attributes. Since current editing approaches lack mechanisms to coordinate within-subject edits, they produce conflicts and degraded performance (Duan et al., 2025).

Full fine-tuning (FT) unsurprisingly achieves high accuracy, since all parameters are supervised, but its locality performance is consistently poor. This weakness is particularly evident on PeaCoK-Ex, where each subject contains only a single fact and the model tends to overfit to that fact, yielding worse locality compared to PerInfoKG. FT-Circuit, which tunes only a small portion of parameters, also achieves high accuracy, but its performance exhibits large variance across models and datasets, suggesting that circuit-only supervision is unstable and requires more principled methods. LoRA yields reasonably strong performance overall, demonstrating its robustness as a lightweight alternative. ROME performs competitively on PeaCoK-Ex, where the setting aligns with its design of editing a single fact per subject, but it degrades substantially on PerInfoKG, where multiple facts for the same subject must be updated jointly. MEMIT-Merge, in principle, should be better suited to multi-fact updates, yet its performance remains suboptimal in our setting. In PerInfoKG, the multiple field values for people are already well-established semantic anchors in the $init$-$LLM$'s embedding space. Averaging these heterogeneous and largely independent value vectors may cause the merged representation to collapse or drift in undesirable ways, which could explain MEMIT-Merge's failure to produce coherent updates in this setting. AlphaEdit achieves strong performance on PeaCoK-Ex dataset, but it still struggles on PerInfoKG, again reflecting the challenge of handling multiple fact updates per subject.

WISE shows robust performance in terms of Retention rate on PeaCoK-Ex dataset, however, it failed to do so on the more complex PerInfoKG dataset. A plausible explanation is that the routing thresholds are not effectively calibrated for the multi-fact-per-subject setting. WISE relies on locality data to make routing decisions; since this data incorporates the initial personal knowledge but excludes general world knowledge, the method effectively preserves Locality but fails to maintain general knowledge. ICE performs reasonably well on PeaCoK-Ex, which is a comparatively simpler single-fact setting. However, its performance drops substantially on PerInfoKG. As noted in Huang et al. (2025), although ICE generally maintains strong locality and retention, they can vary substantially depending on the domain of the edited knowledge and the LLM itself. Since PerInfoKG contains many heterogeneous relations spanning diverse personal-information fields, the updates may vary across multiple semantic domains, which could plausibly explain the degraded locality and retention.

In contrast, our method consistently delivers both high accuracy and strong locality across datasets, showing stable performance with low variance and outperforming all baselines.

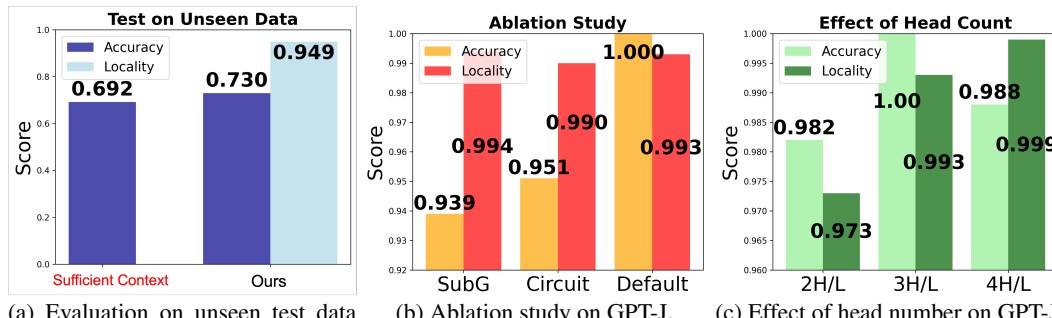

(a) Evaluation on unseen test data using GPT-J.    (b) Ablation study on GPT-J.    (c) Effect of head number on GPT-J.

Figure 3: (a) Evaluation of our align module against sufficient context on GPT-J using the test split of PerInfoKG data. (b) Ablation study on GPT-J with PeaCoK-Ex: SubG denotes the variant without subgraph representations ($\mathbf{h}_{txt}^{\mathcal{G}(h)}, \mathbf{h}_{kge}^{\mathcal{G}(h)}$), and Circuit denotes the variant using *low-importance heads*. (c) Experiments varying the number of heads per layer when applying knowledge injection.

## 5.4 EXTENDED EXPERIMENTAL RESULTS

### 5.4.1 UNSEEN TRIPLES

To assess the generalization ability of our approach, we test whether the align module can apply updates to triples outside its training supervision. The idea is that once trained on multiple (initial, updated) triple pairs, it should be able to inject new updates into the LLM even for unseen triples. Using the dataset split in Section 3.2, where for each of the 2,000 individuals we partition the 23 fields into 17 mutable attributes (for editing) and 6 immutable attributes (for evaluating locality), the module is trained on the resulting 32,952 update instances in the train split and evaluated on 500 unseen instances each in the validation and test splits. As shown in Figure 3a, our method achieves 73% accuracy with 94.9% locality, demonstrating strong generalization. Notably, this accuracy surpasses the sufficient context scenario (Joren et al., 2025), which scored 0.692 accuracy under 100% retrieval success, showing that our approach incorporates new knowledge effectively without direct supervision while preserving prior knowledge.

### 5.4.2 CONTRIBUTION ANALYSIS

**Ablation Study.** We evaluated the effect of circuits by replacing important heads with low-importance ones (i.e., heads with the lowest importance scores across layers), and assessed the contribution of subgraph representations by removing the subgraph features from both KG and LLM (Figure 3b). Both ablations led to performance degradation, with the subgraph removal causing the larger drop. This indicates that subgraph information captures higher-order structure beyond triples, while circuit-level steering also contributes to effective personal information update.

**Head Count.** We next varied the number of heads per layer that constitute the circuit. Using three or four heads yields better performance than using only two, although four does not consistently outperform three (Figure 3c). Interestingly, locality degrades when the circuit contains only two heads. A possible explanation is that, with fewer heads in the circuit, the parameter update for each head increases, causing each steering vector to grow larger in magnitude compared to the three- or four-head cases, which in turn amplifies unintended side effects, particularly the reduction of locality.

## 6 CONCLUSION

In this work, we introduced a new setting for LLM personalization, where mutable personal knowledge in knowledge graphs must be reflected in the model. We defined a fact-level personalization task and proposed a circuit-level steering method that, unlike finetuning or editing approaches reliant on pre-cached corpora, integrates updates while preserving unchanged personal facts. Our experiments show strong performance, demonstrating effective personalization with minimal forgetting.

## ETHICS STATEMENT

All authors have read and will adhere to the ICLR Code of Ethics. Our experiments use only synthetic personal-knowledge datasets (PeaCoK-Ex and PerInfoKG), comprising fictitious individuals and knowledge graph triples; no real human-subject or personally identifiable data were collected, and IRB approval was not applicable. Our method (SPIKE) internalizes updates by steering a sparse set of attention heads rather than retrieving external records, which reduces exposure of raw records but does not by itself ensure legal compliance; any deployment with real data should include consent, access control, auditing, and revocation mechanisms. We note possible dual-use risks, such as injecting false personal facts, and potential biases inherited from backbone LLMs; conflicts of interest and funding sources will be disclosed through the conference system.

## REPRODUCIBILITY STATEMENT

We will release code in `https://anonymous.4open.science/r/SPIKE-F4B6/readme.md`, configuration files, and Dockerized environments to reproduce all results, along with datasets. The repository will include the full training pipeline to create the init-LLM, evaluation scripts for Accuracy, Locality, and the Total Score. Default hyperparameters are reported in Appendix A.8. Hardware and environment details will be documented, and all code, data, and prompts will be released under a permissive license.

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

# A APPENDIX

## A.1 LLM USAGE

We used a large language model (LLM) solely as a writing assistant. Its role was strictly limited to checking grammar, word choice, and stylistic consistency in the manuscript. All aspects of research ideation, experimental design, analysis, and substantive content generation were carried out independently by the authors.

## A.2 TASK FORMULATION & METRICS

We measure the success of knowledge injection using an accuracy metric on the our method and baselines. Specifically, for each updated triple, accuracy is defined as whether all tokens of the gold tail entity are contained in the model's generated output, and we report the average over all updated triples. As described in Section 3.2, $\mathcal{T}^{\text{init}} = \{(s_1, r_1, o_1^{\text{init}}), \ldots, (s_n, r_n, o_n^{\text{init}})\}$ denotes the triples in $\mathbf{KG}^{\text{init}}$ and $\mathcal{T}^{\text{upd}} = \{(s_1, r_1, o_1^{\text{upd}}), \ldots, (s_n, r_n, o_n^{\text{upd}})\}$ denotes the triples in $\mathbf{KG}^{\text{upd}}$, where $n$ is the number of personal factual triples. The set of modified pairs of triples is given by $\mathcal{C} = \{((s_i, r_i, o_i^{\text{init}}), (s_i, r_i, o_i^{\text{upd}})) \mid i \in [n], o_i^{\text{init}} \neq o_i^{\text{upd}}\}$, and accuracy is computed with respect to $\mathcal{C}$. Locality is defined analogously to accuracy, except that it is measured on the set of non-modified triples $\mathcal{R} = \{((s_i, r_i, o_i^{\text{init}}), (s_i, r_i, o_i^{\text{upd}})) \mid i \in [n], o_i^{\text{init}} = o_i^{\text{upd}}\}$. Given $|\mathcal{C}| = m$ and $|\mathcal{R}| = (n - m)$, locality assesses whether the model continues to correctly reproduce the gold tail entities for facts that remain unchanged after the update.

For PeaCoK-Ex, $\mathbf{KG}^{\text{upd}}$ is obtained by modifying $20\%$ of the person–occupation pairs while keeping other attributes fixed. For PerInfoKG, we take a simpler setup: each individual has 23 fields, of which 17 mutable attributes are updated to form $\mathbf{KG}^{\text{upd}}$, while the remaining 6 immutable attributes are left unchanged and used to evaluate locality.

## A.3 DATASET CONSTRUCTION & STATISTICS

PeaCoK Gao et al. (2023), which itself extends commonsense KGs such as ATOMIC Sap et al. (2019), provides a rich set of relations describing personal attributes (`experience`, `routine_habit`, `characteristics`, `goal_plan`) along with their social counterparts (`relationship_experience`, `relationship_routine_habit`, `relationship_characteristics`, `relationship_goal_plan`). However, the original PeaCoK graph does not explicitly contain person entities or individualized personal information. To construct a personalized knowledge graph suitable for temporal update evaluation, we develop **PeaCoK-Ex**, an extended personal-knowledge version of PeaCoK, following a three-stage transformation pipeline.

**(1) Refinement of the Raw PeaCoK KG.** The construction process begins by extracting profession-related information from subjects in the raw PeaCoK KG, many of which contain natural-language descriptions such as "I am a X who ..." or "I am a Y."

*Profession Extraction.* Regular-expression patterns are applied to identify embedded job descriptions. Subjects matching: "I am a {profession} who ...", or "I am a {profession}" are mapped to their canonical profession label, resulting in a set of unique professions and a mapping from verbose subject strings to standardized profession entities.

*Knowledge Graph Reformatting.* Each triple whose subject contains a profession description is rewritten by replacing the natural-language subject with its extracted profession label. Triples in which the rewritten subject and object collide (e.g., profession = object), or cases where the object is itself a profession, are removed to avoid semantic inconsistencies. The outcome is a cleaned, profession-centric KG that provides a consistent schema for downstream construction.

**(2) Introduction of Synthetic Persons and Person–Profession Assignment.** To convert the profession-centric KG into a personalized one, we introduce synthetic individuals.

*Person Entity Generation.* For each extracted profession, one synthetic person entity (e.g., *Frances Travis*) is created, producing as many individuals as there are professions (822 persons in total).

*Deterministic Person–Profession Linking.* Each synthetic person is deterministically linked to a unique profession via the triple:

$$\langle \text{Person}_i, \text{ has\_a\_job\_of}, \text{ Profession}_j \rangle.$$

This guarantees one-to-one person–profession assignments and converts the graph into a person-centric structure encoding explicit occupational information.

**(3) Reverse-Relation Augmentation.** To support bidirectional reasoning, the final stage augments the KG with reverse relations.

*Reverse Relation Construction.* For each relation $r$, a reverse relation $r^{-1}$ is defined. Every triple $(h, r, t)$ is expanded into both $(h, r, t)$ and $(t, r^{-1}, h)$. Examples include:

- has\_a\_job\_of $\rightarrow$ is\_a\_job\_of
- characteristic $\rightarrow$ is\_a\_characteristic\_of
- is\_experience\_of $\rightarrow$ has\_experience\_of

This approximately doubles the triple count and ensures that relational information is navigable in both directions.

**Final Dataset.** After applying the pipeline, PeaCoK-Ex contains 822 synthetic person entities, each associated with exactly one profession, yielding 1,644 person–job triples (including reverse relations), along with a large number of job-related attribute triples inherited from the PeaCoK schema. The resulting KG includes 105,258 triples, 49,818 entities, and 18 relation types (counting reverse relations). Table 3 summarizes the key statistics of the dataset.

We treat this graph as the initial snapshot $\text{KG}^{\text{init}}$. To simulate temporal evolution, we generate $\text{KG}^{\text{upd}}$ by modifying 20% of the person–occupation pairs while keeping all other attributes unchanged. This establishes a realistic temporal-update evaluation setting where only a portion of personal information changes over time.

**PerInfoKG** is a synthetic dataset constructed over 2,000 fictitious individuals and 23 personal information fields. Each field and its corresponding candidate and possible probability weight are shown in Table 9. The probability weight is determined based on real-world statistics. For each individual, we partition the 23 fields into 17 *mutable* attributes used for editing and 6 *immutable* attributes reserved for evaluating locality, ensuring that every individual contributes to both edit and locality evaluation. Importantly, to reflect a real-world setting, we blocked some cases, such as a subject changing the education level from 'PhD' into 'middle school'. The dataset contains 2,134

Table 3: Statistics of the extended PeaCoK-Ex dataset.

| | |
|---|---|
| #Entities | 49,818 |
| #Relations | 18 (including reverse) |
| #Triples | 105,258 |
| #Synthetic person entities | 822 |
| #Person–job triples | 1,644 |
| Update ratio | 20% of person–occupation pairs |

entities and 46,000 triples at $KG^{init}$, with 33,952 update instances used to derive $KG^{upd}$. We split these instances into 32,952 for training and 500 each for validation and test. This split is designed to evaluate the model's capacity to generalize to *unseen updated triples*, i.e., new user-specific facts that were not observed during training, thereby testing the adaptability of our alignment module.

To provide a more complete description of how PerInfoKG is constructed, we now detail the underlying two-stage generation pipeline comprising (1) initial profile generation and (2) rule-based temporal updates.

**(1) Initial Profile Generation ($KG^{init}$)**

*Attribute Space and Sampling Distributions.*

Each field is accompanied by a categorical value set and, when applicable, a weight vector reflecting realistic demographic tendencies.

*Name List Construction.* We prepare a list of 2,000 unique names, which serve as identifiers for each fictitious individual.

*Consistent Sampling of Education, Major, and Job.* To enforce logical coherence across dependent attributes:

- Education level is sampled first.
- If the education level corresponds to higher education (bachelor's, master's, or PhD), a major is sampled from non-`None_MAJ` categories; otherwise, the major is fixed to `None_MAJ`.
- Given the assigned major, the job is sampled from the predefined mapping `major_to_jobs`, which lists occupations compatible with each major.

*Sampling of Remaining Attributes.* All remaining fields (e.g., `religion`, `address`, `political_affiliation`, `hobby`, `medical_conditions`, `marital_status`, `drinking_frequency`) are independently sampled based on their categorical distributions. The resulting profiles form the initial snapshot $KG^{init}$.

**(2) Rule-Based Temporal Updates ($KG^{upd}$).**

*Mutable Fields.* A designated set of 17 mutable fields is defined: `address`, `phone_model`, `pets_owned`, `medical_conditions`, `education_level`, `hobby`, `political_affiliation`, `job`, `housing_type`, `commuting_means`, `exercise_frequency`, `major`, `favorite_food`, `favorite_music`, `diet_type`, `marital_status`, `drinking_frequency`. All other fields (i.e., `name`, `sex`, `nationality`, `blood_type`, `race_ethnicity`, `age_group`) remain fixed across time (the value of `age_group` represents birth-year groupings (e.g., "1990s"), and is therefore treated as immutable.).

*Education-Level Progression.* Education level is updated according to:

- If not already PhD, the level is advanced to the next tier.
- If the prior major was `None_MAJ` but the updated level enters higher-education tiers, a new major is sampled from the non-`None_MAJ` set.

*Job Update Constrained by Updated Major and Education.* Given the updated major and education levels:

- Candidate jobs are retrieved from `major_to_jobs[major]`.
- Degree requirements are enforced (e.g., "scientist" requires a master's degree; "professor" requires a PhD).
- A new job is selected, distinct from the previous one.

*Updates for All Other Mutable Fields.* For each remaining mutable field, a new value is uniformly sampled from the remaining options excluding original value.

Because each updated profile is generated deterministically from its initial version—with logical constraints, controlled randomness, and aligned field dependencies—$KG^{init}$ and $KG^{upd}$ together

Table 4: Top-8 circuit similarity analysis across datasets and models.

| Dataset Pair | GPT-J | Qwen |
|---|---|---|
| PeaCoK-Ex vs. PerInfoKG | **0.6242** | **0.4754** |
| PeaCoK-Ex vs. Known1000 | 0.5774 | 0.4375 |
| PerInfoKG vs. Known1000 | 0.5335 | 0.4100 |

Table 5: Selective Finetuning Results

| Model | Target | Ratio (%) | Acc. (%) |
|---|---|---|---|
| GPT2-XL | FFN | 4.3 | 27.47 |
| | Heads | 4.6 | 99.75 |
| Llama3.1-8B | FFN | 2.2 | 20.83 |
| | Heads | 2.1 | 89.24 |

form a clean two-time-step benchmark suitable for evaluating temporal personal-information updates, locality preservation, and knowledge-update behavior in LLMs.

### A.4 VISUALIZATION

In this section, we visualize and analyze the circuits identified in Section 4.1. Figure 4 presents the min–max normalized importance scores of attention heads across layers for each LLM (Wang & Komatsuzaki, 2021; Yang et al., 2024) and personal information dataset (PeaCoK-Ex, PerInfoKG).

### A.5 CIRCUIT ANALYSIS FOR PERSONAL INFORMATION LOCALIZATION (EXTENDED RESULTS)

In Section 4.1, we report selective finetuning results on Qwen2.5-7B-Instruct, where updating attention heads yields higher accuracy than updating FFNs under the same training budget. To examine whether this behavior generalizes beyond Qwen, we apply the same experimental setup used in Section 4.1 to GPT2-XL and Llama3.1-8B. As shown in Table 5, finetuning attention heads consistently outperforms finetuning FFNs for both models under comparable cost.

For recent architectures such as Llama and Qwen, updating even a single FFN layer can easily drive accuracy close to 100% when the finetuning budget is not constrained. To make the comparison between FFN and head updates meaningful, we therefore fix the total number of training tokens processed during finetuning and evaluate accuracy under this matched budget. Under this setting, attention-head updates remain more effective than FFN updates across all models considered.

### A.6 CIRCUIT STRUCTURE ANALYSIS

Overall, models of the same architecture exhibit highly consistent circuit structures across personal information datasets. Interestingly, Qwen shows a clear and consistent pattern: head 11 dominates in the early layers, heads 2 and 3 are salient in layers 4–11, head 20 becomes prominent in deeper layers, and importance gradually dissipates toward the final layers. In contrast, GPT-J does not display dominance of specific heads but instead distributes importance across multiple heads within each layer, suggesting a more diffuse circuit structure for handling personal data.

**Top-$k$ Circuit Similarity Analysis.** To further examine the specialization of circuits for personal information processing, we conduct a top-$k$ analysis that focuses only on the most critical attention heads within each layer. Specifically, for each layer $\ell$, we identify the $k$ highest-scoring heads based on importance scores and construct a binary mask $\mathbf{M}_\ell \in \{0, 1\}^H$, where $H$ is the number of heads. The masked importance matrix is then obtained as $\mathbf{S}^{(k)} = \mathbf{S} \odot \mathbf{M}$, where $\mathbf{S}$ is the original importance

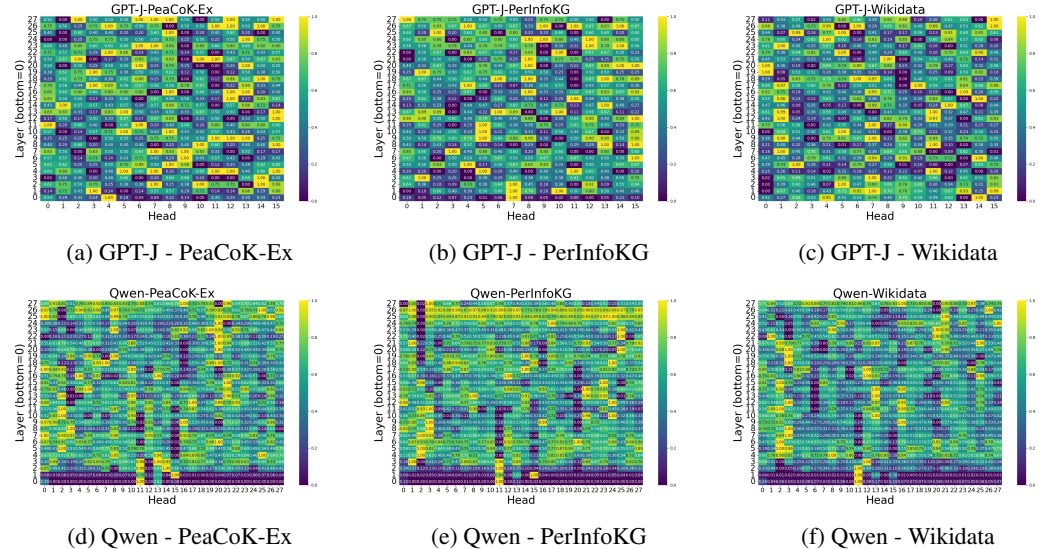

(a) GPT-J - PeaCoK-Ex      (b) GPT-J - PerInfoKG      (c) GPT-J - Wikidata

(d) Qwen - PeaCoK-Ex      (e) Qwen - PerInfoKG      (f) Qwen - Wikidata

Figure 4: Heatmap visualizations of important components identified in GPT-J (Wang & Komatsuzaki, 2021) and Qwen (Yang et al., 2024) across PeaCoK-Ex, PerInfoKG, and Wikidata (Meng et al., 2022) (*Known1000*). Each plot shows attention heads (x-axis) by layers (y-axis). The similarity is highest when comparing personal knowledge datasets.

score matrix and $\odot$ denotes element-wise multiplication. Circuit similarity is then measured as the cosine similarity between the flattened masked matrices.

Table 4 presents pairwise cosine similarities with $k=8$ (top-8 heads per layer). The results reveal a clear pattern: personal information datasets (PeaCoK-Ex and PerInfoKG) consistently show higher similarity to each other than to general Wiki-based knowledge data (Known1000).

For GPT-J, the similarity between PeaCoK-Ex and PerInfoKG reaches $0.6242$, notably higher than the cross-domain similarities of $0.5774$ (PeaCoK-Ex vs. Known1000) and $0.5335$ (PerInfoKG vs. Known1000). Qwen shows the same trend, with $0.4754$ (PeaCoK-Ex vs. PerInfoKG) exceeding $0.4375$ and $0.4100$ for cross-domain pairs.

**Implications for Personal Information Processing.** These findings provide strong evidence for the existence of specialized neural circuits for personal information processing in large language models. The consistently higher intra-domain similarity (personal vs. personal) compared to cross-domain similarity (personal vs. general) suggests that LLMs recruit distinct computational pathways for handling personal information queries. This specialization has important implications for both interpretability and privacy: identifying dedicated personal circuits enables targeted interventions such as selective parameter editing or circuit-level privacy protection. Moreover, the contrast between GPT-J's diffuse attention patterns and Qwen's concentrated head dominance indicates that circuit specialization strategies may differ across architectures, highlighting an avenue for future research.

**Global or Personal circuit.** We compare "global" and "personal" circuits in terms of effectiveness (by comparing accuracy and locality) and scalability. First, we compared the accuracy and locality in the PeaCoK-Ex dataset when applying "global" or "personal" circuits. For applying personal circuit, it achieves 100% accuracy and 99.3% locality. Compared to the performance of global circuit in Table 2 (100% accuracy, 99.3% locality), the difference between the two settings is negligible, indicating that both approaches are similarly effective in terms of accuracy and locality.

We also conducted the same comparison on PerInfoKG using GPT-J. With global circuit steering, the model achieved 94.38 accuracy, 96.34 locality, and a total score of 95.35. Using personal (user-specific) circuits yielded 93.03 accuracy, 96.26 locality, and a total score of 94.62. As in PeaCoK-Ex,

the performance gap between the two remains minimal, further confirming that personal circuits do not offer a meaningful advantage in effectiveness.

The distinction becomes more pronounced when considering scalability. "Personal" circuits require computing a new circuit for each incoming user before SPIKE can be applied, which incurs ongoing per-user overhead. In contrast, the global circuit can be maintained and reused as new users arrive. Its robustness to unseen users is further supported by the results demonstrated in Figure 3(a), where the global circuit remains effective even under an unseen test scenario.

In summary, while both circuit types exhibit comparable effectiveness, the global circuit offers substantially better scalability. For this reason, we adopt the global circuit in our method.

### A.7 ADDITIONAL EXPERIMENT

**Evaluation on Unstructured and Noisy Entity Variants**  The personal-KG datasets used in our main experiments employ normalized entity strings for consistency, where descriptive modifiers and free-form expressions are removed. Since personal information in natural settings can include such descriptive or unstructured forms, we also construct a variant of PeaCoK-Ex that preserves the original entity strings from the source corpus and evaluate SPIKE under this extended condition.

The original PeaCoK dataset includes rich, free-form descriptions within entity tokens (e.g., `(aids in the completion of large projects, is a social routine or habit of, heavy duty equipment operator who work hard at my job), (Arlo Hill, has a job of, heavy-duty equipment operator who work long and hard))`.

Table 6: Evaluation on Peacok-Ex-Noisy

| Model | Base | Acc. | Loc. | Tot. |
|-------|------|------|------|------|
| GPT-J | Ours | 98.17 | 93.47 | 95.76 |
| | AlphaEdit | 6.10 | 96.40 | 11.47 |
| | Finetune | 95.12 | 9.42 | 17.14 |

Such expressions contain descriptive modifiers and unstructured phrasing (e.g., "who work long and hard") that introduce linguistic variability not present in the canonical entity labels. In PeaCoK-Ex, we intentionally removed this variability to isolate the underlying structured entity (e.g., heavy-duty equipment operator). To evaluate robustness under more realistic conditions, we constructed PeaCoK-Ex-Noisy, which retains all original descriptive, unstructured, and diverse entity strings while preserving the same number of people. Importantly, both the input triple and the updated target triple use these full noisy expressions. Thus, the model must perform personalization and update reasoning without relying on normalized labels, instead handling full naturalistic variation. All other experimental settings remain identical to those used in the main evaluation.

### A.8 HYPERPARAMETER SETTING

As shown in Eq. 4, our objective consists of three components: the negative log-likelihood term $\mathcal{L}_{\text{NLL}}$ that enforces the updated KG to be reflected in the initial LLM, the KL divergence term $\mathcal{L}_{\text{KL}}$ that preserves knowledge unrelated to the updates, and the norm-based penalty $\mathcal{L}_{\text{norm}}$ that prevents the steering vector from deviating excessively from the LLM representation. The additional terms are controlled by the hyperparameters $\lambda_1, \lambda_2 \in \{0.0, 0.1, \ldots, 0.5\}$ to find the optimal configuration. Moreover, we treat the number of intervened attention heads $k$ as a hyperparameter, setting $k = 2$ for the PeaCoK-Ex dataset and $k = 3$ for the PerInfoKG dataset. The best-performing settings are $\lambda_1 = 0.1, \lambda_2 = 0.2$ on the PeaCoK-Ex dataset and $\lambda_1 = 0.0, \lambda_2 = 0.0$ on the PerInfoKG dataset, consistently across LLM backbones.

### A.9 LLM PERSONALIZATION

Wang et al. (2024b) proposed EMG-RAG, which addresses personalized question answering by extracting personal memories from smartphone conversations and app screenshots. Their approach introduces an Editable Memory Graph (EMG) that supports dynamic memory operations, including insertion, deletion, and replacement of personal information. The system employs reinforcement learning to train an agent for adaptive memory selection, moving beyond fixed Top-K retrieval methods to handle complex queries requiring diverse memory combinations. While their work focuses

Table 7: Two case studies illustrating model responses to an occupation-related question and a multi-hop reasoning question drawn from the PeaCoK-Ex dataset.

| | Case Study 1 | Case Study 2 |
|---|---|---|
| Subject | Natalie King | Lucian Newman |
| Initial Occupation | Head of Corporation | Popular President |
| Updated Occupation | Guitarist | Chess Player |
| Occupation Question | Natalie King has a job of | Lucian Newman has a job of |
| Answer | Guitarist | Chess Player |
| Multi-hop Question | Natalie King has a job whose characteristic is | Lucian Newman has a job that requires |
| Answer | skilled in musical performance | strategy and good decision-making skills |

on memory retrieval and selection for downstream tasks such as QA and form autofill, our approach tackles a different challenge: efficiently incorporating updated knowledge graph information into LLM behavior without full model retraining.

A line of research (Prahlad et al., 2025) has explored personalization approaches for LLMs by structuring personal data from applications such as calendars, conversational chats, and emails into knowledge graphs for smart response generation. Their approach leverages RAG with smaller models to provide factually correct responses using dynamically updated KGs, addressing privacy concerns by keeping sensitive data locally rather than sending it to cloud-based LLM providers. The system focuses on using KG-based retrieval to enhance LLM responses for personal queries and smart reply generation. However, this work focuses on retrieval-based personalization rather than updating LLM knowledge as personal information changes.

A relevant baseline is KGT Sun et al. (2024), which adapts to evolving user information by directly modifying knowledge graph, such as adding or removing triples based on user feedback. During inference, it relies on retrieving these updated triples and appending them to the input context, thereby depending on the model's in-context reasoning capabilities to incorporate the external information. Unlike KGT, which modifies the external input context via retrieval, our approach operates directly on the model's internal state by injecting steering vectors into activations, effectively shifting the model's internal processing toward the new information without altering the input prompt.

## A.10 CASE STUDY

Table 8: A case study illustrating model responses to behavior personalization drawn from the PEA-COK dataset.

| | Original LLM (GPT-J) | Original LLM (GPT-J) + SPIKE |
|---|---|---|
| Subject | Natalie King | Natalie King |
| Occupation | Head of Corporation | Guitarist |
| Behavior Question | Based on the job information Natalie King has, how will Natalie King respond to the following question? "What does 'leading a successful performance' mean to you?" | |
| Answer | "I would define it as the ability to influence **others** to achieve success." | "I would define it as having a successful performance **myself**." |

This section presents two case studies that demonstrate the effectiveness of SPIKE: the first assesses its ability to handle multi-hop questions involving updated facts, and the second examines whether the LLM exhibits personalized behavioral tendencies.

Table 7 summarizes two examples drawn from PeaCoK-Ex, focusing on the subjects 'Natalie King' and 'Lucian Newman'. We assess the performance of SPIKE under two types of questions: (i) a direct query regarding the updated occupation, and (ii) a multi-hop query that requires reasoning based on the occupation. In both cases, SPIKE successfully guides the language model to provide accurate and contextually appropriate responses. Specifically, when asked about the characteristics of Natalie King's job, the steered model generated "skilled in musical performance," which aligns closely with the updated occupation of Guitarist rather than the previous role as Head of Corporation. Similarly, in response to a query about Lucian Newman's job requirements, the model produced "strategy and good decision-making skills," reflecting the essential competencies of a Chess Player. These results demonstrate that SPIKE effectively steers the model not only to answer direct occupation queries but also to generate coherent and realistic responses in multi-hop reasoning scenarios. Consequently, this case study reinforces the applicability of SPIKE in both direct and reasoning-based evaluation settings.

Table 8 examines whether SPIKE enables the LLM to adjust its behavior in accordance with an updated fact. The table presents a case study involving a tone-validation question, demonstrating that the model's response varies depending on the updated occupation. For instance, when analyzing the term 'successful performance', the interpretation of 'performance' shifts significantly depending on whether the occupation is 'head of a corporation' or updated to 'guitarist'. The results indicate that, without incorporating SPIKE, the model's responses remain anchored to the initial occupation, emphasizing organizational roles rather than individual characteristics, as a 'head of a corporation' must account for all employees. In contrast, when SPIKE is applied, the model shifts toward a more individualized interpretation, well reflecting the occupation into that of a (solo) guitarist.

Table 9: Defined Fields and Probability Weights for 23 Candidate Fields in PerInfoKG.

| Field | Candidate List | Probability List |
|---|---|---|
| City | ['NewYork', 'Toronto', 'Berlin', 'Seoul', 'Tokyo', 'Paris', 'Sydney'] | [0.24, 0.17, 0.12, 0.06, 0.15, 0.13, 0.13] |
| Alcohol Frequency | ['infrequently', 'sometimes', 'often', 'socially'] | [0.2, 0.3, 0.3, 0.2] |
| Favorite Food | ['ramen', 'pizza', 'sushi', 'pasta', 'bibimbap', 'steak', 'burger'] | [0.1429, 0.1429, 0.1429, 0.1429, 0.1429, 0.1429, 0.1429] |
| Music Genre | ['Pop', 'Jazz', 'Classical', 'Hip-Hop', 'Rock', 'Indie', 'Electronic'] | [0.1429, 0.1429, 0.1429, 0.1429, 0.1429, 0.1429, 0.1429] |
| Diet Type | ['vegan', 'vegetarian', 'omnivore', 'halal'] | [0.1, 0.1, 0.75, 0.05] |
| Nationality | ['Korean', 'American', 'Japanese', 'British', 'Canadian'] | [0.05, 0.4, 0.15, 0.2, 0.2] |
| Housing | ['apartment', 'house', 'dormitory', 'studio', 'villa'] | [0.3, 0.3, 0.1, 0.1, 0.2] |
| Commute | ['subway', 'bus', 'car', 'bike', 'walking'] | [0.3, 0.3, 0.3, 0.05, 0.05] |
| Marital Status | ['single', 'married'] | [0.45, 0.55] |
| Exercise Frequency | ['rarely', 'infrequently', 'weekly', 'daily'] | [0.15, 0.5, 0.3, 0.05] |
| Blood Type | ['A', 'B', 'AB', 'O'] | [0.34, 0.27, 0.11, 0.28] |
| Religion | ['Christianity', 'Buddhism', 'Islam', 'Hinduism', 'Atheism'] | [0.3, 0.07, 0.24, 0.15, 0.24] |
| Hobby | ['reading', 'swimming', 'painting', 'gaming', 'hiking', 'cycling', 'traveling'] | [0.1429, 0.1429, 0.1429, 0.1429, 0.1429, 0.1429, 0.1429] |
| Gender | ['male', 'female'] | [0.5, 0.5] |
| Phone Model | ['iphone', 'galaxy', 'mi', 'pixel'] | [0.4, 0.4, 0.1, 0.1] |
| Race/Ethnicity | ['White', 'African American', 'Asian', 'Hispanic', 'other'] | [0.2, 0.2, 0.1, 0.2, 0.3] |
| Age Group | ['1940s', '1950s', '1960s', '1970s', '1980s', '1990s', '2000s'] | [0.04, 0.06, 0.1, 0.12, 0.17, 0.24, 0.27] |
| Medical Condition | ['None_MED', 'diabetes', 'hypertension', 'asthma', 'depression', 'arthritis', 'allergies'] | [0.55, 0.07, 0.08, 0.06, 0.05, 0.05, 0.1] |
| Political Affiliation | ['Democrat', 'Republican', 'Independent', 'Unaffiliated'] | [0.2, 0.3, 0.05, 0.45] |
| Pet | ['None_PET', 'dog', 'cat', 'other'] | [0.4, 0.3, 0.2, 0.1] |
| Education Level | ['middle school', 'high school', 'bachelor's degree', 'master's degree', 'PhD'] | [0.1, 0.57, 0.25, 0.06, 0.02] |
| Major | ['Computer Science', 'Business', 'Biology', 'Mechanical Engineering', 'Economics', 'English Literature', 'Nursing', 'None_MAJ'] | [0.125, 0.125, 0.125, 0.125, 0.125, 0.125, 0.125, 0.125] |
| Job | ['accountant', 'artist', 'barista', 'cashier', 'counselor', 'data analyst', 'dentist', 'developer', 'doctor', 'driver', 'economist', 'engineer', 'entrepreneur', 'manager', 'nurse', 'pilot', 'politician', 'professor', 'researcher', 'scientist', 'soldier', 'teacher', 'writer'] | [0.0637, 0.0179, 0.0179, 0.0179, 0.0208, 0.0156, 0.0156, 0.0312, 0.0156, 0.0179, 0.025, 0.0156, 0.1701, 0.1284, 0.0417, 0.0156, 0.0543, 0.1314, 0.0677, 0.0469, 0.0179, 0.0335, 0.0179] |

