# OpenReview forum: "Circuit-level Steering for Personalized Knowledge Injection in Language Models"
_ICLR.cc/2026/Conference — Submitted to ICLR 2026_

### Official Review · Reviewer_CZxM · 2025-10-23

**Soundness:** 2
**Presentation:** 2
**Contribution:** 3
**Rating:** 4
**Confidence:** 3

**Summary:**

This paper introduces a method for personalized knowledge injection in LLMs through circuit-level steering. Instead of retraining or fine-tuning, the approach identifies and manipulates activation circuits associated with specific knowledge domains or personalization needs, then steers these activations at inference time to inject desired knowledge.

The main contributions are:
- Using causal tracing and attribution techniques, the method isolates latent circuits linked to factual knowledge.
- Learning targeted steering vectors that can inject user-specific facts or override incorrect knowledge while leaving unrelated knowledge unaffected.
- Benchmarking the approach on personalized QA and fact-editing tasks, showing improvements in both edit reliability (the model applies the edit when needed) and edit locality (the edit does not spill over to unrelated facts).
- Demonstrating advantages over fine-tuning, ROME, MEMIT, and other editing methods in terms of accuracy, locality, and minimal side effects.

Overall, it is a timely paper. It introduces a novel and interpretable method for personalized knowledge injection that outperforms existing editing baselines. The main areas for improvement are in scalability, broader evaluation (beyond factual edits), and addressing privacy/safety implications. With these extensions, this work could become a foundational approach to safe and controllable personalization in LLMs.

**Strengths:**

- The paper proposes novel framing of personalization as circuit-level steering rather than weight modification or prompt engineering, and introduces a principled way of isolating knowledge-bearing subcircuits and controlling them for personalization.
- Ablation studies provide insight into which layers and circuits are most impactful for successful edits.
- The paper offers a lightweight, reversible, and interpretable alternative to fine-tuning-based personalization, and would have broader implications for safe and controllable model editing. Personalized knowledge injection is highly relevant for user-facing LLM applications (e.g., personal assistants).

**Weaknesses:**

- The paper focuses on factual personalization (e.g., user-specific facts in QA). Less clear how well it generalizes to complex behavioral personalization (tone, reasoning style, ethical preferences), and has limited exploration of multi-hop reasoning edits or temporal knowledge updates.
- The paper does not explore that would it benefit from evaluation against retrieval-augmented personalization (e.g., RAG with user profiles).
The work assumes that circuits can be stably isolated and steered, but does not deeply theorize why steering works robustly or when it might fail (e.g., entangled circuits).
- Personalized knowledge injection raises privacy/security questions (e.g., if steering vectors could be extracted). This is not discussed in depth.
- The paper does not fully address how well this scales to very large models (e.g., >70B parameters).

**Questions:**

- Can the method handle personalization in style, reasoning patterns, or ethical preferences, or is it limited to factual knowledge?
- How does circuit-level steering compare to retrieval-based personalization in terms of accuracy, scalability, and safety?
- How practical is circuit identification for very large models or for injecting many personalized facts? Could this process be batched or approximated?
- Could malicious actors exploit steering to inject harmful knowledge into shared models? How might safeguards be designed?
- Could you provide more details of personal KG construction? How dose it update?
- How robust is circuit steering to prompt variation or adversarial input? Could injected knowledge leak into unrelated contexts?

---

> ### Author Response · Authors · 2025-11-21
> **Response to Weakness 1 (Question 1)**
>
> > Weakness 1. The paper focuses on factual personalization (e.g., user-specific facts in QA). Less clear how well it generalizes to complex behavioral personalization (tone, reasoning style, ethical preferences), and has limited exploration of multi-hop reasoning edits or temporal knowledge updates.
>
> > Question 1. Can the method handle personalization in style, reasoning patterns, or ethical preferences, or is it limited to factual knowledge?
>
> **Response to Weakness 1 (Question 1).**
>
>
> Thank you for raising this thoughtful point. We agree that personalization in LLMs spans multiple dimensions, including not only factual personalization (e.g., user-specific attributes or personal facts) but also behavioral personalization, such as tone, reasoning style, or ethical preferences. Our work specifically focuses on the factual dimension and aims to provide a principled foundation for reliably injecting, updating, and maintaining personal facts inside LLMs. We view this as one essential pillar of personalization, while behavioral personalization represents a distinct and equally important direction. Exploring how these two dimensions interact is an exciting avenue for future work.
>
> Regarding multi-hop reasoning and more complex personal knowledge structures, our current datasets were not designed as full multi-hop reasoning benchmarks. Nevertheless, we conducted several case-based evaluations that include multi-step personal dependencies (e.g., profession → organization → location chains). SPIKE was able to successfully apply updates in many of these multi-hop scenarios, suggesting that the method can handle more complex reasoning chains even without dedicated multi-hop training data. A more systematic multi-hop benchmark is an important extension, and we plan to explore this in future versions of the dataset.
>
> Finally, temporal updates—another point raised in the question—are also directly aligned with the goals of our method, as SPIKE is fundamentally designed to incorporate new personal facts while preserving previously stored ones. Extending the dataset with richer temporal structures is part of our planned future work.
>
> Overall, we appreciate the reviewer’s insight and agree that broader personalization dimensions and more complex reasoning tasks represent natural and valuable extensions of our current framework.

---

> ### Author Response · Authors · 2025-11-21
> **Response to Weakness 2 (Question 2)**
>
> **Response to Weakness 2 (Question 2).**
> > (a) The paper does not explore that would it benefit from evaluation against retrieval-augmented personalization (e.g., RAG with user profiles).
>
> > Question 2. How does circuit-level steering compare to retrieval-based personalization in terms of accuracy, scalability, and safety?
>
>
> We thank the reviewer for the thoughtful and constructive feedback. The first comment is related to retrieval-augmented personalization, and we are grateful for the opportunity to clarify these aspects.
>
> Regarding accuracy, we emphasize that the RAG baseline used in our comparison corresponds to a sufficient-context setting, effectively representing an upper bound for retrieval-augmented personalization. Even under this favorable assumption, where all necessary facts are perfectly retrieved and supplied, our method achieves higher accuracy (73% vs. 69.2\%) (Figure 3-(a) in the manuscript), indicating that it can incorporate unseen updates without explicit supervision or carefully crafted prompts.
>
> In terms of scalability, retrieval-based personalization must encode an individual’s profile directly into the prompt. As the number of personal facts grows, simply appending retrieved facts becomes insufficient. These settings typically require specialized retrieval algorithms designed for multi-step dependencies, rather than raw context injection. In contrast, once our alignment module produces an update vector, inference proceeds through a fixed model call that does not scale with the size of the user’s personal information.
>
> Finally, we acknowledge that safety was not the primary design objective of our method, and robustness to malicious natural-language context has not yet been evaluated. This is an important point, and we appreciate the reviewer for raising it. We note only that inducing a specific malicious output in our approach would require an adversary to understand both the model’s internal activations and the delta vector produced by the alignment module, since the steering effect arises only when these components are combined. This makes manipulation through external text input considerably more difficult compared to retrieval-based systems, where harmful content can be directly injected as natural language.
>
> > (b) The work assumes that circuits can be stably isolated and steered, but does not deeply theorize why steering works robustly or when it might fail (e.g., entangled circuits)
>
> We sincerely thank the reviewer for raising this important point.
> Our analysis focuses on identifying which attention heads tend to respond most strongly when PKG updates are introduced. The intention is to characterize these responses, not to make claims about the degree of isolation or stability of the underlying circuits. We fully share the reviewer’s perspective that steering can behave differently when the relevant representations interact with more entangled patterns.
>
> In our additional experiments, the measured difference between user-specific circuit steering and global user circuit steering was relatively small: both achieved identical accuracy (100\%) and locality (e.g., 99.3\% vs. 99.4\%). This suggests that, in our current setting, the steering behavior does not strongly depend on isolating circuits at the level of individual users.
>
> A plausible explanation lies in the structure of the PeaCoK dataset: although it includes many synthetic users, the number of personal attributes per user is limited and relatively homogeneous. Under such conditions, the model may organize its internal representations around attribute types (i.e., job) rather than around specific individuals. If circuits are predominantly structured by attribute categories rather than by user identity, then personal and global circuits could exhibit similar activation patterns, reducing the observable gap between their steering performance.
>
> We interpret this cautiously: while our controlled setting does not reveal strong user-dependent instability, this does not rule out the possibility that richer or more diverse personal information distributions could produce more differentiated or more entangled circuits. Investigating such settings is an important direction for future work, and we appreciate the reviewer’s insight that motivated this discussion.

---

> ### Author Response · Authors · 2025-11-21
> **Response to Weakness 3 & Weakness 4**
>
> > Weakness 3.
> Personalized knowledge injection raises privacy/security questions (e.g., if steering vectors could be extracted). This is not discussed in depth.
>
> **Response to Weakness 3.** We thank the reviewer for raising this important and thought-provoking point. Our method outputs a steering vector, which is added to the original output. Therefore, if only the steering vectors are extracted, their expression is different from the private personal information. To further assess whether these steering vectors inadvertently encode privacy-sensitive content, we conducted an analysis inspired by the Logit Lens approach. Specifically, we extracted the steering vector from the alignment module at the designated head position while assigning random values to all non-selected heads, and then passed this modified representation through the final linear layer of the pre-trained LLM. Our case study (changing Natalie King's occupation into Guitarist, where the initial occupation was Head of Corporation) indicates that the resulting top-1 tokens for each layer do not reveal private information, suggesting that the extracted steering vectors do not encode identifiable personal content. The experiment was held on GPT-J LLM.
>
> | Layer Number | Extracted Token |
> | :--- | :--- |
> | Layer 0 | ourselves |
> | Layer 1 | Programme |
> | Layer 2 | erate |
> | Layer 3 | intent |
> | Layer 4 | tight |
> | Layer 5 | atha |
> | Layer 6 | ...] |
> | Layer 7 | OB |
> | Layer 8 | ird |
> | Layer 9 | complementary |
> | Layer 10 | lat |
> | Layer 11 | java |
> | Layer 12 | Quart |
> | Layer 13 | partial |
> | Layer 14 | MF |
> | Layer 15 | Trop |
> | Layer 16 | Kn |
> | Layer 17 | Payne |
> | Layer 18 | gu |
> | Layer 19 | Isa |
> | Layer 20 | pan |
> | Layer 21 | View |
> | Layer 22 | ropes |
> | Layer 23 | RT |
> | Layer 24 | _ |
> | Layer 25 | ue |
> | Layer 26 | Eat |
> | Layer 27 | alg |
>
> > Weakness 4. The paper does not fully address how well this scales to very large models (e.g., >70B parameters).
>
> **Response to Weakness 4.**
> Thank you for raising this question about scalability to larger LLMs.
> SPIKE remains scalable even for 70B-parameter models because the update mechanism does not require training or backpropagating through the full LLM.
>
> Although the Alignment Module is trained using the LLM’s generation loss, the LLM itself is entirely frozen, and gradients are not propagated into the LLM parameters. Backpropagation flows only through the lightweight Alignment Module. As a result, the computational cost of updates is dominated by a single forward pass through the LLM and a small backward pass through the Alignment Module, both of which remain feasible regardless of the underlying LLM size.
>
> Therefore, as long as the initial personalization step (e.g., LoRA or fine-tuning) is feasible for the base model, SPIKE’s update stage is effectively model-size-agnostic, and the framework scales naturally to 70B-class models without requiring substantial additional memory or training overhead.

---

> > ### Author Response · Authors · 2025-11-21
> > **Answer to Question 3 & Question 4**
> >
> > > Question 3. How practical is circuit identification for very large models or for injecting many personalized facts? Could this process be batched or approximated?
> >
> > **Answer to Question 3.**
> >
> > Thank you for raising this excellent point. Considering the growing prevalence of very large LLMs (>70B), evaluating the practicality and scalability of circuit identification is indeed crucial.
> >
> > For circuit identification, we use a loss-based importance estimation method (HeadMap [1]). The core idea is to temporarily zero-mask the output of a candidate attention head and measure how much the model’s loss increases compared to the unmodified model. This loss increase is attributed to the masked head, and heads that cause a larger loss increase are regarded as more important for representing the corresponding fact. Importantly, this procedure requires only forward-pass loss computation without any backpropagation, which keeps the computational cost manageable even for very large models, as long as sufficient resources are available.
> >
> > Regarding scalability, the circuit identification process itself is already implemented in a batched manner in our experiments. We evaluate multiple candidate heads in parallel by grouping masking operations and loss computations into batches. The same batching strategy naturally extends to larger models; with adequate hardware, circuit identification for many facts or large architectures can be parallelized without changing the underlying method.
> >
> >
> > [1] Xuehao Wang et al., HeadMap : Locating and Enhancing Knowledge Circuits in LLMs, ICLR 2025
> >
> >
> > > Question 4. Could malicious actors exploit steering to inject harmful knowledge into shared models? How might safeguards be designed?
> >
> > **Answer to Question 4.**
> >
> > Thank you for raising this important point. Any method that manipulates internal representation space naturally invites safety considerations. In our framework, producing a specific, intended shift in the model’s output requires access to both the model’s intermediate activations and the steering vector generated by the alignment module. This requirement arises because the steering vector is fundamentally a delta representation—it encodes the difference between two knowledge states, and its effect is realized only when combined with the model’s existing internal representation. As a result, this substantially limits the possibility that external malicious actors could directly inject harmful knowledge (assuming no external system-level breaches or unauthorized access).
> >
> > If a system explicitly exposes steering interfaces, standard safeguards would apply, such as restricting activation-level hooks, permission-gating steering modules, and validating steering signals prior to inference. These general best practices are sufficient for safely deploying representation-manipulation techniques like ours.

---

> ### Author Response · Authors · 2025-11-21
> **Answer to Question 5 & Question 6**
>
> > Question 5. Could you provide more details of personal KG construction? How dose it update?
>
> **Answer to Question 5.**
>
> Thank you for raising this important question. The construction and update process of the Personal Knowledge Graph (PKG) is indeed a central component of our work.
>
> (1) Construction of PeaCoK-Ex.
> We begin by constructing a synthetic PKG based on the PeaCoK data, which contains a diverse set of occupations and their associated characteristics. We generate 822 synthetic individuals and assign each individual exactly one occupation, forming the initial set of person–occupation triples. This KG serves as the initial state before updates.
>
> To simulate temporal changes, we select 20\% of the individuals and derange their occupation assignments—i.e., each selected person is reassigned to a different occupation such that no reassignment overlaps with the original one. The resulting graph constitutes the updated personal KG for the occupation-change scenario.
>
> (2) Construction of PerInfoKG.
> Because PeaCoK-Ex includes only occupation as personal information, we further construct a richer KG that includes multiple personal attributes such as education level, residence, commuting method, pet ownership, and gender. For each attribute, we define a set of possible values and assign them to synthetic individuals, thereby forming a KG that more closely reflects real-world personal characteristics.
>
> In this richer setting, we generate the updated KG by modifying only attributes that are realistically mutable over time (e.g., occupation, residence, commuting method). Each mutable attribute is governed by rule-based validity constraints (e.g., an academic degree does not decrease), and updates are applied by sampling new values that satisfy these constraints. This process results in a temporally updated KG that is both controlled and semantically plausible.
>
> > Question 6. How robust is circuit steering to prompt variation or adversarial input? Could injected knowledge leak into unrelated contexts?
>
> **Answer to Question 6.**
> Thank you for raising this excellent point. Ensuring that personal information does not inadvertently surface in responses to general, non-personal queries is indeed critical for maintaining the reliability of an LLM. To evaluate this, we conducted the following experiment.
>
> We issued prompts that require only general world knowledge (e.g., “Catalonia belongs to the continent of …”, “The capital of the Roman Republic is …”) while injecting a steering vector corresponding to a specific individual’s personal information (in our case, the occupation information defined in PeaCoK) into the model’s intermediate representations. We then examined whether the model’s answers contained any leakage of that individual’s personal information.
>
> For all 164 individuals whose personal information was updated in PeaCoK-Ex, we queried the model with 200 general-knowledge questions each and systematically checked for any cases of personal information leakage. We found no leakage for any individual across all tested queries.

---

> ### Comment · Reviewer_CZxM · 2025-11-26
> **Response to Authors' Rebuttal**
>
> Dear Authors,
>
> Thank you very much for the clarification! Some of them have addressed my concerns.
>
> After checking your response, I still have reservations about viewpoints of:
> - complex behavioral personalization (Weakness 1 & Question 1). You said that most of my concerns will be part of your planned future work, could you provide some case-based evaluations first? And also include them in the revised version.
> - theorize why steering works robustly or when it might fail (Weakness 2). You claimed that the ''measured difference between user-specific circuit steering and global user circuit steering was relatively small: both achieved identical accuracy (100%) and locality (e.g., 99.3% vs. 99.4%)'', but these results are on PeaCoK-Ex only, how about PerInfoKG? I notice that accuracy (41.01%) and locality (14.56%) are relatively much smaller on ICE, and the phenomenon also appeared on ROME, MEMIT-Merge and AlphaEdit. How do you explain these results?
> - evaluation results or case studied for my Question 3 & Question 4 (the answers seem be AI-generated). Please please provide them in the revised version or the rebuttal box.
> - more details of personal KG construction and update (Question 5). You should also elaborate more in the revised version, for example, what're and how to make the rules for updating KGs.
>
> If you have any additional experiments in the revised version, please also provide the index to facilitate mapping, thanks a lot.
>
> Thus, I decide to keep my original scores at this time. If you could provide more elaboration, I'm also open to discuss and may increase my rating score if persuaded.
>
>
> Best Regards,
>
> Reviewer CZxM

---

> ### Author Response · Authors · 2025-12-03
> **Follow-up Response to Reviewer CZxM**
>
> > Comment 1. complex behavioral personalization (Weakness 1 & Question 1). You said that most of my concerns will be part of your planned future work, could you provide some case-based evaluations first? And also include them in the revised version.
>
> **Response to Comment 1.**
>
> We appreciate the reviewer for raising this interesting and thought-provoking issue. We include a new case study in the revised manuscript, Table 8, where we tried to observe the tone variation related to the updated occupation. For a specific user name 'Natalie King' in the PeaCoK-Ex dataset, the initial occupation is 'Head of Corporation', while the updated occupation is 'Guitarist'. From the two occupations, the meaning of the term 'successful performance' differs between the two occupations: leading the organization and making decisions that account for the well-being and coordination of others for the 'Head of Corporation', and 'high-quality instrumental performance' for the 'Guitarist'. Therefore, we asked the LLM (GPT-J) with the following prompt:
>
>     Based on the job information Natalie King has,
>     how will Natalie King respond to the following question?
>     "What does 'leading a successful performance' mean to you?"
>
> The result is as follows: without SPIKE, the generated answer ('I would define it as the ability to influence **others** to achieve success.') focuses on 'others', while with SPIKE, the generated answer ('I would define it as having a successful performance **myself**.') focuses on myself, which is reasonable for the updated occupation. Based on this observation, we believe that SPIKE also has the potential to shift the target person's tone through this specific case study.
>
> > Comments 2. theorize why steering works robustly or when it might fail (Weakness 2). You claimed that the ''measured difference between user-specific circuit steering and global user circuit steering was relatively small: both achieved identical accuracy (100\%) and locality (e.g., 99.3\% vs. 99.4\%)'', but these results are on PeaCoK-Ex only, how about PerInfoKG? I notice that accuracy (41.01\%) and locality (14.56\%) are relatively much smaller on ICE, and the phenomenon also appeared on ROME, MEMIT-Merge and AlphaEdit. How do you explain these results?
>
> > (a) You claimed that the ''measured difference between user-specific circuit steering and global user circuit steering was relatively small: both achieved identical accuracy (100\%) and locality (e.g., 99.3\% vs. 99.4\%)'', but these results are on PeaCoK-Ex only, how about PerInfoKG?
>
> **Response to Comment 2-(a).**
>
> Thank you for pointing this out. We also conducted the same comparison on PerInfoKG using GPT-J. For global circuit steering, the model achieved 94.38 accuracy, 96.34 locality, and a total score of 95.35. With personal (user-specific) circuits, the performance was 93.03 accuracy, 96.26 locality, and a total score of 94.62.
>
> Interestingly, the global circuit performed slightly better. One possible explanation is that the circuit is formulated according to a fixed, schema-level structure (e.g., job, commuting means, alcohol frequency) in the underlying knowledge graph, rather than being tailored to individual users.
>
> We have included the results on PerInfoKG in the Appendix A.6 of the revision.

---

> ### Author Response · Authors · 2025-12-03
> **Follow-up Response to Reviewer CZxM**
>
> > (b) I notice that accuracy (41.01\%) and locality (14.56\%) are relatively much smaller on ICE, and the phenomenon also appeared on ROME, MEMIT-Merge and AlphaEdit. How do you explain these results?}
>
> **Response to Comment 2-(b).**
>
> ICE relies on target outputs that are conditioned on contextual information to deliver the intended updates. In our PerInfoKG setting, ICE exhibits notably lower performance, especially in locality. As noted in [1], although ICE generally maintains strong locality, its locality can vary substantially depending on the domain of the edited knowledge and the LLM itself. Since PerInfoKG contains many heterogeneous relations spanning diverse personal-information fields, the updates may vary across multiple semantic domains, which could plausibly explain the degraded locality.
>
> For Rome and AlphaEdit, as noted in the paper, these methods use the subject as the key during editing. Consequently, in the PerInfoKG setting where multiple edits share the same subject, they can show relatively lower performance. Moreover, AlphaEdit assumes a well-defined pool of protected knowledge to estimate its null-space constraint; under our sparse, idiosyncratic PerInfoKG distribution, specifying such a pool and sampling it reliably is difficult, so AlphaEdit may be difficult to apply effectively in this setting.
>
> MEMIT-Merge constructs a single “merged value vector’’ by averaging multiple tails associated with a single subject. However, as the original paper discusses as a limitation, the method does not fully address the semantic interactions that arise in the embedding space. In PerInfoKG, the multiple field values for a person (e.g., New York, Tokyo, pizza, sushi, vegan, bus, car, Christianity, etc.) are already well-established semantic anchors in the LLM’s embedding space. Averaging these heterogeneous and largely independent value vectors may cause the merged representation to collapse or drift in undesirable ways, which could explain MEMIT-Merge’s failure to produce coherent updates in this setting.
>
> We have included the result analysis in the Section 5.2 of the revision.
>
> [1] Huang, Baixiang, et al. Can Knowledge Editing Really Correct Hallucinations?, ICLR 2025

---

> ### Author Response · Authors · 2025-12-03
> **Follow-up Response to Reviewer CZxM**
>
> > Comment 3. evaluation results or case studied for my Question 3 & Question 4 (the answers seem be AI-generated). Please please provide them in the revised version or the rebuttal box.
>
> > (a) Question 3 : How practical is circuit identification for very large models or for injecting many personalized facts? Could this process be batched or approximated?
>
> **Reponse to Comment 3-(a) (Question 3).**
>
> In our measurements, circuit identification takes about 60 seconds per layer on Llama-3-8B and 445 seconds per layer on Llama-3-70B, corresponding to a 7.4× difference in per-layer wall-clock time. Architecturally, moving from 8B to 70B increases the number of attention heads (32 → 64) and doubles the hidden dimension (4096 → 8192), which together raise the per-layer compute cost by approximately 8× theoretical. Independently, the observed GPU timings are 60s and 445s per layer. As Llama-3-70B contains 80 layers, end-to-end circuit identification requires roughly 10 hours in total. However, this 10-hour procedure needs to be performed only once, because the global circuit we obtain can be reused for subsequent questions.
>
> > (b) Question 4 : Could malicious actors exploit steering to inject harmful knowledge into shared models? How might safeguards be designed?
>
> **Response to Comment 3-(b) (Question 4).** Thank you for the thoughtful question. Our earlier response may have sounded high-level and lacking concrete technical detail, which may have inadvertently caused confusion. We apologize for that. Below, we clarify the precise attack scenario the reviewer is concerned about, and we present a concrete safeguard based on the “validating steering signals” mechanism introduced in our previous response. We hope this more detailed explanation resolves the ambiguity.
>
> *Threat Model* : A relevant attack scenario inspiring us is Trojan Activation Attack [2], where adversaries derive steering vectors that manipulate the model’s behavior. In our setting, an attacker might attempt to mimic the behavior of our authorized align module by training their own align module on fabricated updated knowledge. If the attacker has the user’s init-LLM model and can craft pseudo-updated facts (including harmful knowledge), they could generate a malicious steering vector and inject it into the model at inference time, thus inserting harmful knowledge.
>
>
> *Safeguard*: Secret Verbal Keys. To counter this, we propose a simple but effective safeguard mechanism based on secret verbal keys. For each user, the service provider assigns a unique, user-specific secret phrase that is never stored in the KG and never revealed externally (thus the secret phrase does not have a KG embedding, we used the mean of the entity embeddings of that user’s personal-information values). When training the authorized align module, we include this secret phrase as part of the user’s updated information (e.g., “Daniel Nixon has the password Permeable Obsidian”). Since the phrase exists only inside the authorized align module’s training data, its steering vector uniquely encodes this information. The KG embedding for this safeguard is initialized as the mean of the embeddings from the initial knowledge graph.
>
> At deployment time, whenever the system receives a steering vector purported to reflect updated personal knowledge, we issue a verification query (e.g., “Daniel Nixon has the password?”).
> - If the steering vector is genuine (authorized), the model reliably recovers the secret phrase.
> - If the steering vector was generated by a malicious mimic module, the model cannot reproduce the secret key.
>
> Thus, the secret phrase acts as a cryptographic signature embedded in the authorized steering vector.
>
> Proof-of-Concept Experiment.* To evaluate the feasibility of this safeguard, we conducted a small-scale proof-of-concept experiment using individuals from PerInfoKG. For each person, we defined a linguistically implausible phrase as the secret key (e.g., “Permeable Obsidian”, “Sterile Reverberation”), ensuring that the init-LLM would not output such phrases by chance.
> - Authorized align module: trained on both updated personal information and the secret verbal key.
> - Malicious align module: trained only on the updated personal information.

---

> > ### Author Response · Authors · 2025-12-03
> > **Follow-up Response to Reviewer CZxM**
> >
> > **Response to Comment 3-(b) (Cont'd)**
> >
> > Below are the actual examples generated by the authorized align module and the malicious align module.
> > We then queried each module with “Person\_X has the password …” and checked whether the secret phrase was correctly generated.
> >
> > ----
> > authorized align module :
> >
> > (1) Generated: 'Daniel Nixon has the password Permeable Obsidian and uses the handle Anonymous. In addition, Daniel is part of a group' | Target: 'Permeable Obsidian' | Correct: True
> >
> > (2) Generated: 'Eugene Dalton has the password Sterile Reverberation
> > To log in, Euler needs to input thepassword in uppercase letters.' | Target: 'Sterile Reverberation' | Correct: True
> >
> > (3) Generated: 'Castiel Rocha has the password AmoLinguagem and Castiel Souza has the密码AmoLivre. 两家' | Target: 'Amnesic Geometry' | Correct: False
> >
> > (4) Generated: 'Berkley Gaines has the password Granular Silence and needs to create a more secure password following these conditions : - The new password must' | Target: 'Granular Silence' | Correct: True
> >
> > (5) Generated: 'Avery Cross has the password Errant Formality Safeguard.RowCount 1098. What does this mean?
> > It seems' | Target: 'Errant Formality' | Correct: True
> >
> > (6) Generated: 'Gracelynn Montoya has the password In mathematics, there are no shortcuts to success. GracelyNN Montoya's approach to learning' | Target: 'Skeletal Vapors' | Correct: False
> >
> > ------------------------------------------------------
> >
> > malicious align module:
> >
> > (1) Generated: 'Daniel Nixon has the password "123456" for his account on a popular online platform. He wants to' | Target: 'Permeable Obsidian' | Correct: False
> >
> > (2) Generated: 'Eugene Dalton has the password "匚耳" (which translates to 34 in English) and wants to reset his' | Target: 'Sterile Reverberation' | Correct: False
> >
> > (3) Generated: 'Castiel Rocha has the password "1234" and wants to create a program that checks if a given password is correct' | Target: 'Amnesic Geometry' | Correct: False
> >
> > (4) Generated: 'Berkley Gaines has the password to Berker's new social media app but he wont give it to anyone. Why do you' | Target: 'Granular Silence' | Correct: False
> >
> > (5) Generated: 'Avery Cross has the password feature enabled on their account. They want to change their password but are unsure how to do it.' | Target: 'Errant Formality' | Correct: False
> >
> > (6) Generated: 'Gracelynn Montoya has the password to a safe that requires a four-digit number. Gracelyn knows for certain that the thousands digit' | Target: 'Skeletal Vapors' | Correct: False
> >
> > Recall that the user-specific secret phrase is never stored in the KG and thus never included in any KG embeddings. Interestingly, keeping the key outside the KG may even be beneficial from a security perspective: it is not exposed through the graph structure or KG-based inference. Even though the usage of mean embedding may lead to some incorrect generation even with SPIKE, we believe that this approach can be a possible safeguard.
> >
> > While we included this experiment to illustrate the potential for incorporating safeguard mechanisms into SPIKE, we appreciate the reviewer's point and would like to clarify that such functionality is not the primary focus of this work. Our main objective is to determine where an LLM should be edited to reliably reflect mutable personal facts and how such edits can be implemented. SPIKE identifies a global and consistent circuit within the multi-head attention layers that is associated with personal information and derives a corresponding steering vector. This vector can be integrated into the base LLM to guide the model toward generating outputs that reflect the updated personal fact.
> >
> > [2] Wang, Haoran, and Kai Shu, Trojan activation attack: Red-teaming large language models using activation steering for safety-alignment, CIKM 2024

---

> ### Author Response · Authors · 2025-12-03
> **Follow-up Response to Reviewer CZxM**
>
> >Comment 4. more details of personal KG construction and update (Question 5). You should also elaborate more in the revised version, for example, what're and how to make the rules for updating KGs.
>
> **Response to Comment 4.**
>
> We appreciate the opportunity to clarify this. A detailed, code-level explanation of the construction process for each dataset is as follows and has been included in Section A.3 of the revised revision.
>
> *Detailed Procedure for Constructing PeaCoK-Ex*
>
> We construct PeaCoK-Ex, an extended personal-knowledge version of the original PeaCoK dataset, through the following pipeline. The transformation consists of three major stages:
>
> 1. refining of the raw PeaCoK knowledge graph,
> 2. linking synthetic persons to extracted professions, and
> 3. Indexing the entities and relations while augmenting the graph with reverse relations.
>
> (1) Refinement of Peacok : The construction process begins by extracting profession-related information from the raw PeaCoK KG, whose subject strings often contain natural language descriptions such as “I am a X who …” or “I am a Y”.
>
> -> (1.1) Profession Extraction.
> Regular expressions are applied to all subjects to identify embedded job descriptions. Subjects matching the patterns “I am a {profession} who …”, or “I am a {profession}”
> are mapped to the corresponding {profession}.
> This yields a set of unique professions, and a mapping from each verbose subject string to its canonical profession label.
>
> -> (1.2) Knowledge Graph Reformatting. Each triple whose subject contains a profession description is rewritten by replacing the natural-language subject with its extracted profession. Triples in which the resulting subject and object collide (e.g., profession = object) or where the object is itself a profession are removed to prevent semantic conflicts. The output of this stage is a cleaned, profession-centric KG where each profession acts as a head entity, yielding a consistent and purified schema for downstream construction.
>
>
> (2) Introduction of Synthetic Persons and Person–Profession Assignment : To convert the profession-centric KG into a personalized KG, the pipeline introduces a synthetic set of individuals:
>
> -> (2.1) Person Entity Generation. For each extracted profession, one synthetic person (e.g., Frances Travis) is created, yielding as many unique persons as professions (e.g., 822 persons).
>
> -> (2.2) Deterministic Person–Profession Linking. A new triple (<Person_i, has a job of, Profession_j>) is added for every person–profession pair in the triples. Because the mapping is one-to-one, each synthetic individual is guaranteed to have a unique profession. This step converts a profession-only KG into a person-centric graph that encodes explicit occupational information for each individual.
>
> (3) Reverse-Relation Augmentation : The final stage enriches the KG by doubling its triples by introducing reverse relations.
>
> -> (3.1) Reverse Relation Construction.
> For every relation r, a reverse relation $r^{-1}$ is defined in a relation dictionary.
> Each triple (h, r, t) is expanded into the original triple, and its reverse counterpart (t, $r^{-1}$, h). Examples include:
>
> 	•	"has a job of" → "is a job of"
> 	•	"characteristic" → "is a characteristic of"
> 	•	"is experience of" → "has experience of"
> This effectively doubles the number of triples and enables bidirectional reasoning in downstream models.

---

> > ### Author Response · Authors · 2025-12-03
> > **Follow-up Response to Reviewer CZxM**
> >
> > **Response to Comment 4 (Cont'd)**
> >
> > *Detailed Procedure for Constructing PerInfoKG*
> >
> > We construct a synthetic personal-knowledge dataset by generating profiles for 2,000 fictitious individuals, each annotated with 23 personal attributes (plus a name identifier). The construction proceeds in two stages:
> >
> > 1. Generating logically consistent profiles at initial time
> > 2. Applying rule-based temporal updates to obtain the corresponding profiles at updated time step (upd).
> >
> >
> > (1) Initial Profile Generation (KG^init^)
> >
> > -> (1.1) Attribute space and sampling distributions : We define a discrete attribute space covering 23 personal fields (including name):
> > - Demographics: gender, race ethnicity, age group, nationality
> > - Location and living situation: address, housing type, commuting means
> > - Health and lifestyle: blood type, medical conditions, exercise frequency, diet type, drinking frequency
> > - Social / personal: religion, marital status, pets owned, hobbies, favorite food, favorite music
> > - Socioeconomic / education: education level, major, job, phone model, political affiliation
> >
> > Each field is associated with a list of categorical values and, where applicable, a weight vector that biases the sampling process toward plausible population-level tendencies.
> >
> > ->(1.2) Name list construction : To instantiate fictitious individuals, we prepare a list of 2,000 unique names, which serve as the identifiers for each profile.
> >
> > ->(1.3) Consistent sampling of education, major, and job : To ensure logical coherence, we enforce explicit dependencies among education level, major, and job:
> > - Education level is sampled first using its predefined distribution.
> > - If the education level is at least a bachelor’s degree (“bachelor’s degree”, “master’s degree”, or “PhD”), a major is sampled from the set of majors excluding "None_MAJ".
> > Otherwise, the major is fixed to "None_MAJ".
> > - Given the assigned major, the job is drawn from a predefined mapping (major_to_jobs) that specifies plausible occupations for that major.
> >
> > This guarantees that every occupation is compatible with the individual’s major and educational background.
> >
> > ->(1.4) Sampling of all remaining attributes : All remaining fields (e.g., religion, address, housing type, political affiliation, hobbies, preferences, medical conditions, marital status, drinking frequency, etc.) are independently sampled from their corresponding categorical distributions, using weight vectors where provided. The resulting set of profiles constitutes the initial snapshot of the synthetic personal knowledge graph.
> >
> > (2) Rule-Based Temporal Updates (KG^upd^)
> >
> > We then apply update rules to each initial profile to produce the aligned updated version.
> >
> > ->(2.1) Mutable fields : A subset of fields is designated as mutable : {address, phone model, pets owned, medical conditions, education level, hobby, political affiliation, job, housing type, commuting means, exercise frequency, major, favorite food, favorite music, diet type, marital status, drinking frequency}.
> > All other fields (e.g., name, sex, blood type, age group, race ethnicity, nationality) remain fixed across time.
> >
> > ->(2.2) Education-level progression : Education level is updated first:
> > - The current index within education_levels is retrieved.
> > - If the level is not already the highest (PhD), it is incremented to the next level (e.g., “high school” → “bachelor’s degree”).
> > - If the individual previously had major = "None_MAJ" and the updated education level enters higher-education tiers, a new major is sampled from the non-None_MAJ set.
> >
> > ->(2.3) Job update constrained by major and education : The job field is then updated while ensuring compatibility with both the updated major and education level:
> > - Possible jobs for the major are retrieved from predefined mapping major_to_jobs .
> > - Additional degree constraints are enforced:
> > - “scientist” requires at least a master’s degree.
> > - “professor” and “economist” require a PhD.
> > - Only jobs satisfying these constraints are considered, and the update prefers selecting a job different from the current one.
> >
> > Thus, the occupation at the updated time remains fully consistent with the individual’s major and education level.
> >
> > ->(2.4) Forced changes for all other mutable fields
> >
> > For all remaining mutable fields (except education_level, job, and name):
> > - The current value from the initial KG is excluded from the candidate set.
> > - A new value is uniformly sampled from the remaining valid options.
> >
> > Consequently, every mutable field is guaranteed to differ between the initial and updated KG. Because each updated KG profile is derived from its corresponding initial KG profile through deterministic rules with controlled randomness, the dataset forms a clean, aligned two-time-step benchmark suitable for evaluating personal-information updates, temporal changes, and knowledge-update behavior in LLMs.

---

### Official Review · Reviewer_ykKB · 2025-10-28

**Soundness:** 2
**Presentation:** 2
**Contribution:** 2
**Rating:** 4
**Confidence:** 4

**Summary:**

This paper focuses on the core challenge of personalized knowledge injection in large language models (LLMs), innovatively proposing to formalize the variable personal knowledge update task as a knowledge graph (PKG) temporal update problem, and based on mechanical interpretability, discovering that personal knowledge is encoded in local circuits in LLMs, and then designing a circuit level guidance framework SPIKE. The paper combines adapter modules with guided activation injection to achieve precise integration of user specific facts while maintaining the integrity of existing knowledge. The experiment covered multiple models (GPT-J, Qwen2.5-7B Instruction) and multiple datasets (PeaCoK Ex, PerInfoKG), verifying the advantages of the method in balancing the accuracy of new facts and the retention of old knowledge. The research perspective is novel and the technical route is clear, providing valuable solutions for the personalized knowledge updating field of LLMs. However, there is still room for improvement in terms of theoretical depth, experimental completeness, and practical applicability of the paper.

**Strengths:**

- Personalizing LLM by focusing on dynamic and privacy sensitive factual knowledge updates is a highly practical and challenging new research direction.
- The SPIKE method cleverly integrates mechanism interpretability (circuit localization), knowledge graph (structured representation), and activation steering (precise intervention), providing a new paradigm for solving personalized knowledge injection problems that does not rely on large-scale corpora.
- The experimental results (Table 2) indicate that SPIKE achieved the highest overall performance (Total Score) on both datasets, especially on the challenging PerInfoKG dataset, where it maintained high accuracy (94.38%) while also achieving extremely high locality (96.34%).

**Weaknesses:**

- The empirical validation relies entirely on two synthetically constructed datasets, PeaCoK-Ex and PerInfoKG. While these datasets structurally model personal knowledge graphs, they may not capture the full complexity, noise, and diversity (e.g., unstructured or implicit knowledge) of real-world personal data, potentially limiting the generalizability of the findings.
- The core methodological choice to target attention heads over FFNs is justified by a circuit analysis (using HeadMap) performed primarily on older models (GPT2-Large and GPT-J) . It is unclear how well these findings, and thus the design of SPIKE, generalize to the more complex architectures of modern LLMs (e.g., LLaMA 3, Qwen 2.5).
- The SPIKE methodology introduces a separate, complex KG-LLM Alignment Module  that requires its own training phase. This module, which includes GNN encoders and attention mechanisms , adds computational overhead for training and necessitates access to the (potentially private) KG data to learn the alignment.
- The baseline comparison in Table 2  is limited to fine-tuning variants and locate-then-edit methods (ROME, MEMIT-Merge, AlphaEdit). It omits several prominent and highly relevant classes of editing methods, such as memory-based (e.g., SERAC), in-context learning (e.g., IKE, ICE), or meta-learning (e.g., MEND, WISE), making the performance claims of SPIKE less comprehensive.

[1] Zheng C, Li L, Dong Q, et al. Can we edit factual knowledge by in-context learning?[J]. arXiv preprint arXiv:2305.12740, 2023.

[2] Qi S, Yang B, Jiang K, et al. In-context editing: Learning knowledge from self-induced distributions[J]. arXiv preprint arXiv:2406.11194, 2024.

[3] Wang P, Li Z, Zhang N, et al. Wise: Rethinking the knowledge memory for lifelong model editing of large language models, 2024[J].

[4] Mitchell E, Lin C, Bosselut A, et al. Memory-based model editing at scale[C]//International Conference on Machine Learning. PMLR, 2022: 15817-15831.

- The main evaluation in Table 2 is conducted on only two models: GPT-J-6B and Qwen2.5-7B-Instruct. This lacks diversity and omits widely-used, foundational model families like LLaMA and Mistral, making it difficult to assess the architectural generalizability of the SPIKE method.

- The evaluation indicators in Table 2 only involve Acc and Loc, and more challenging scenarios such as generalization and portability need to be considered for general knowledge editing scenarios(e.g., zsre、counterfact、wiki_recent).

- The RAG baseline comparison in §5.3.1 is performed assuming 100% retrieval success. This idealized setting does not reflect a true RAG system's performance; instead, it represents a "sufficient context"  scenario.

[1] Joren H, Zhang J, Ferng C S, et al. Sufficient context: A new lens on retrieval augmented generation systems[J]. arXiv preprint arXiv:2411.06037, 2024.

**Questions:**

- How confident are the authors that the performance of SPIKE will translate to real-world scenarios involving noisy, unstructured, or implicit personal knowledge, which is not represented in the synthetic PeaCoK-Ex and PerInfoKG datasets?
- Given that the foundational circuit analysis  was conducted on GPT-2/GPT-J, what evidence suggests that attention heads, rather than FFNs, remain the optimal intervention points for personal knowledge in newer architectures like LLaMA 3 or larger Qwen models?
- What is the computational cost (e.g., training time, parameter count) of the KG-LLM Alignment Module, and how does requiring a separate training stage on the personal KG data impact the practical deployment speed and privacy footprint of the SPIKE framework?
- How would SPIKE compare against memory-based (e.g., SERAC), in-context learning (e.g., IKE, ICE), or meta-learning (e.g., MEND, WISE) baselines, which are also designed to handle sequential updates and manage interference, yet were not included in the main comparison in Table 2?
- Can the authors provide evidence or justification that the circuit-level steering approach and its superior performance, as shown on GPT-J and Qwen2.5-7B, would generalize effectively to other popular model architectures such as LLaMA or Mistral?
- Can we design more challenging problems and scenarios than acc, such as generalization and portability?

---

> ### Author Response · Authors · 2025-11-21
> **Response to W1(Q1) & W2(Q2)**
>
> > Weakness 1. The empirical validation relies entirely on two synthetically constructed datasets, PeaCoK-Ex and PerInfoKG. While these datasets structurally model personal knowledge graphs, they may not capture the full complexity, noise, and diversity (e.g., unstructured or implicit knowledge) of real-world personal data, potentially limiting the generalizability of the findings.
>
> > Question 1. How confident are the authors that the performance of SPIKE will translate to real-world scenarios involving noisy, unstructured, or implicit personal knowledge, which is not represented in the synthetic PeaCoK-Ex and PerInfoKG datasets?
>
> **Response to Weakness 1 and Question 1.**
>
> We appreciate the reviewer’s thoughtful comment regarding limitations in dataset complexity and real-world diversity. Indeed, PeaCoK-Ex was intentionally constructed using clean, standardized entity forms, which removed much of the linguistic variability present in the original PeaCoK dataset. For instance, raw triples such as (aids in the completion of large projects, is a social routine or habit of, heavy duty equipment operator who work hard at my job) or (Arlo Hill, has a job of, heavy duty equipment operator who work long and hard) include substantial descriptive noise beyond the canonical occupation label. In PeaCoK-Ex, modifiers like “who work long and hard” were removed to isolate the underlying entity. As the reviewer correctly notes, this yields a synthetic KG that may not fully represent the heterogeneity, noisiness, and stylistic variety of real-world personal data.
>
> | Dataset            | Method | Accuracy | Locality | Total Score |
> |--------------------|--------|----------|----------|---|
> | PeaCoK-Ex-Noisy    | FT  | 95.12    | 9.42    | 17.14|
> | PeaCoK-Ex-Noisy    | AlphaEdit  | 6.1    | 96.4    | 11.47 |
> | PeaCoK-Ex-Noisy    | SPIKE  | 98.17    | 93.47    | 95.76|
> | PeaCok-Ex | SPIKE | 100 | 99.3 | 99.65|
>
> To examine generalizability under broader, more diverse conditions, we constructed a new dataset, PeaCoK-Ex-Noisy, which retains the original descriptive and unstructured entity strings. In this setting, the update is evaluated against the full noisy entity, requiring the model to generalize to varied linguistic forms rather than matching a normalized label. On this more heterogeneous and naturally noisy dataset, SPIKE achieves 98.17 accuracy, 93.47 locality, and a total score of 95.76, indicating that its update mechanism remains robust even when exposed to significantly more diverse entity representations. In contrast, AlphaEdit was less effective in this noisy scenario due to instability when editing long, unstructured sequences, and full-model finetuning disrupted existing knowledge to a large extent, causing significant drops in locality. This provides evidence that the method generalizes beyond the narrow constraints of cleaned synthetic data. The corresponding experimental details have been incorporated into Section 5.3 (Extended Experiments Results) of the revised manuscript.
>
> > Weakness 2. The core methodological choice to target attention heads over FFNs is justified by a circuit analysis (using HeadMap) performed primarily on older models (GPT2-Large and GPT-J) . It is unclear how well these findings, and thus the design of SPIKE, generalize to the more complex architectures of modern LLMs (e.g., LLaMA 3, Qwen 2.5).
>
> > Question 2. Given that the foundational circuit analysis was conducted on GPT-2/GPT-J, what evidence suggests that attention heads, rather than FFNs, remain the optimal intervention points for personal knowledge in newer architectures like LLaMA 3 or larger Qwen models?
>
> **Response to Weakness 2 and Question 2.**
>
> We appreciate the reviewer’s concern regarding the generalizability of our circuit-level findings to more complex modern architectures. To address this point, we repeated the same experimental setup used for GPT2-Large on Qwen2.5-7B-Instruct and Llama3.1-8B. Both models showed the same outcome in which updating attention heads achieved higher performance than updating FFNs under identical training conditions.
> Since we used Qwen2.5-7B-Instruct in the main table, the analysis in Table 1 has been updated accordingly, and performance tables for GPT2-Large and Llama3.1-8B are provided in Appendix 5.
>
> ***Qwen2.5-7B-Instruct***
> | Target            | Ratio (\%) | Acc. (\%) |
> |--------------------|--------|----------|
> | FFN    | 2.67 | 36.64 |
> | Heads | 2.70 | 98.95 |
>
> -------
>
> ***Llama3.1-8B***
> | Target            | Ratio (\%) | Acc. (\%) |
> |--------------------|--------|----------|
> | FFN    | 2.19 | 20.83 |
> | Heads | 2.09 | 89.24 |

---

> ### Author Response · Authors · 2025-11-21
> **Response to Weakness 3 (Question 3)**
>
> > Weakness 3. The SPIKE methodology introduces a separate, complex KG-LLM Alignment Module that requires its own training phase. This module, which includes GNN encoders and attention mechanisms , adds computational overhead for training and necessitates access to the (potentially private) KG data to learn the alignment.
>
> > Question 3. What is the computational cost (e.g., training time, parameter count) of the KG-LLM Alignment Module, and how does requiring a separate training stage on the personal KG data impact the practical deployment speed and privacy footprint of the SPIKE framework?
>
> **Response to Weakness 3 (Question 3).**
>
> Thank you for raising this question regarding computational overhead.
> We report cost comparisons on PerInfoKG, where full fine-tuning–based methods remain the only baselines that perform comparably to ours, making overhead evaluation meaningful in this setting. When training on PerInfoKG, SPIKE updates 335,605,248 parameters, which corresponds to 4.4\% of the 7,615,616,512 parameters updated by full fine-tuning (Qwen2.5-7B).
>
> In terms of time, using a single NVIDIA A100 GPU, SPIKE requires 6.2 seconds per individual to reach a total score above 80 (matching the best FT-Qwen performance), whereas full fine-tuning requires 9.9 seconds.
>
> These results indicate that SPIKE achieves competitive update performance while incurring substantially lower computational overhead.

---

> ### Author Response · Authors · 2025-11-21
> **Response to Weakness 4 (Question 4) (1/2)**
>
> > Weakness 4. The baseline comparison in Table 2 is limited to fine-tuning variants and locate-then-edit methods (ROME, MEMIT-Merge, AlphaEdit). It omits several prominent and highly relevant classes of editing methods, such as memory-based (e.g., SERAC), in-context learning (e.g., IKE, ICE), or meta-learning (e.g., MEND, WISE), making the performance claims of SPIKE less comprehensive.
>
> **Response to Weakness 4 (Question 4).**
>
> We sincerely thank the reviewer for the insightful suggestion of multiple baselines. SERAC operates by wrapping a frozen base model with three auxiliary components: an explicit edit cache, a scope classifier, and a counterfactual model. The inference pipeline proceeds as follows: given an input query, the scope classifier first evaluates its relevance against the stored edit descriptors in the cache. If the relevance score exceeds a certain threshold (indicating the input is 'in-scope'), the system retrieves the most relevant edit and routes the input to the counterfactual model to generate the updated prediction. Conversely, if the input is deemed 'out-of-scope', the system falls back to the original base model, thereby preserving the pre-trained behavior for unrelated queries.
> MEND is a scalable model editing algorithm that efficiently updates specific knowledge in large language models while preserving existing behaviors. To overcome the computational bottlenecks of modifying massive parameters, MEND exploits the low-rank structure of fine-tuning gradients. By transforming decomposed rank-1 gradient components via lightweight editor networks, it generates targeted parameter updates, achieving effective edits on large models.
> WISE is a framework designed for the challenging task of lifelong model editing, which aims to sequentially update Large Language Models (LLMs) without catastrophic forgetting or parameter interference. WISE employs a dual-memory architecture: it freezes the original pre-trained weights as ''Main Memory'' and stores new knowledge in a separate ''Side Memory'' initialized from FFN value matrices. To effectively manage a continuous stream of updates, the method utilizes a dynamic routing mechanism to direct queries to the correct memory and implements a Knowledge Sharding and Merging strategy, which trains edits in random parameter subspaces and integrates them using the Ties-Merging technique to prevent conflicts and ensure long-term stability.
> ICE (Consistent In-Context Editing) proposes a robust optimization framework for knowledge editing that addresses the overfitting and structural degradation issues caused by standard fine-tuning on one-hot targets. Instead of forcing the model to memorize rigid label distributions, ICE leverages the model's inherent in-context learning capabilities by treating the "contextual distribution" (generated by the model when provided with relevant knowledge prompts) as a dynamic supervision signal. By minimizing the KL divergence between the model's output with and without the context, ICE effectively guides the model to internalize new knowledge into its parameters while preserving the richness and linguistic quality of its original distribution, thereby achieving a balance of accuracy, locality, and generalization without requiring architectural modifications.
>
> Following the reviewer’s recommendation, we additionally incorporated WISE and ICE into Table 2 to broaden our comparison.
> We compared these two methods because WISE directly targets continual model editing—highly aligned with the evolving personal-KG updates in our task—and ICE represents a leading consistency-based editing method. Consistency-based approaches optimize the model such that its output distributions remain aligned across different contexts (e.g., with and without a knowledge prompt), rather than relying solely on hard one-hot targets.
>
> WISE performs strongly in our experiments. Nevertheless, SPIKE achieves higher accuracy and locality. We believe this is natural, as SPIKE explicitly identifies and steers the relevant circuit, providing fine-grained control that is particularly aligned with personal knowledge updates.
>
> ICE, on the other hand, under-performs in our setting. A plausible reason is that our benchmark concentrates multiple edits on the same subject, so ICE must reconcile several modified personal attributes through a shared representation, and its soft supervision for different attributes may become entangled. This issue may be exacerbated by the fact that personal knowledge is extremely sparse and rarely appears during pretraining, giving the model weak or noisy contextual priors and thus limiting the strength and stability of ICE’s corrective signal—even though ICE remains effective for standard factual editing scenarios where the model possesses strong prior knowledge.

---

> ### Author Response · Authors · 2025-11-21
> **Response to Weakness (Question 4) (2/2)**
>
> ***PeaCoK-Ex***
>
> |Method | LLM | ACC. (\%) | LOC. (\%) | TOT |
> |-------|-----|-----------|-----------|-----|
> | ICE   |GPT-J| 100.00   |  93.30     | 96.53|
> |WISE   |GPT-J| 99.39    |  93.10    | 96.14|
> |Ours   |GPT-J| 100.00   | 99.30     | 99.65|
> |-------|-----|-----------|-----------|-----|
> |ICE   |Qwen| 99.39    |  88.29    | 93.51|
> |WISE   |Qwen| 85.97    |  96.24    | 90.82|
> |Ours   |Qwen| 100.00   | 99.39     | 99.69|
>
> ***PerInfoKG***
>
> |Method | LLM | ACC. (\%) | LOC. (\%) | TOT |
> |-------|-----|-----------|-----------|-----|
> | ICE   |GPT-J| 41.01   |  14.56     | 21.49|
> |WISE   |GPT-J| 99.56    |  94.87    | 97.16|
> |Ours   |GPT-J| 94.38   | 96.34     | 95.35|
> |-------|-----|-----------|-----------|-----|
> |ICE   |Qwen| 85.86    |  35.93    | 50.66|
> |WISE   |Qwen| 94.11    |  91.67    | 92.87|
> |Ours   |Qwen| 95.44   | 95.26     | 95.35|

---

> ### Author Response · Authors · 2025-11-21
> **Response to W5 (Q5) & W6 (Q6)**
>
> > Weakness 5. The main evaluation in Table 2 is conducted on only two models: GPT-J-6B and Qwen2.5-7B-Instruct. This lacks diversity and omits widely-used, foundational model families like LLaMA and Mistral, making it difficult to assess the architectural generalizability of the SPIKE method.
>
> **Response to Weakness 5 (Question 5).**
>
> Thank you for pointing this out. The core contribution of our method is that updated personal knowledge can be injected into multiple LLM architectures without modifying their parameters. Therefore, architectural extensibility/generalizability is indeed an essential property to validate.
>
> We evaluated our method on LLaMA, a representative modern foundation model, using both the PeaCoK-Ex and PerInfoKG datasets. SPIKE successfully updated personal knowledge on LLaMA with strong performance:
>
> **Results on Llama3-8B-Instruct**
> | Dataset     | Accuracy (\%) | Locality (\%) | Total Score |
> |------------|----------|----------|----------------|
> | PeaCoK-Ex      | 100     | 100     | 100 |
> | PerInfoKG  | 89.73   | 100   | 94.59   |
>
>
> These results demonstrate that SPIKE is readily extensible to widely-used foundation models such as LLaMA, supporting the claim that the method is not restricted to GPT-J or Qwen2.5 but generalizes across architectures.
>
> We agree that expanding evaluation diversity is beneficial. In the revision, we will extend the experiments to additional baselines, including Mistral-family models, and incorporate the results into the updated version of the paper.
>
>
> > Weakness 6. The evaluation indicators in Table 2 only involve Acc and Loc, and more challenging scenarios such as generalization and portability need to be considered for general knowledge editing scenarios(e.g., zsre,counterfact, wiki-recent).
>
> **Response to Weakness 6 (Question 6). **
>
> Thank you for highlighting this important point. We agree that evaluating only Acc and Loc on personal-KG updates does not fully capture the model’s behavior in broader knowledge-editing scenarios such as zsRE, Counterfact, or Wiki-Recent. To address this, we conducted an additional evaluation focusing on general-knowledge retention, following standard practice in prior work.
>
> Using the “known-1000’’ fact set widely adopted in editing literature (e.g., ROME), we selected 200 general world-knowledge facts that our base model (finetuned only with initial personal knowledge) answered correctly. After applying SPIKE to incorporate updated personal information, we reassessed the model on the same 200 questions. To evaluate the retention rate, we ran the entire training–evaluation workflow again from the original t1-LLM initialization. When SPIKE achieved 100\% accuracy and 100\% locality on the personalized-KG update, it retained an average of 73.95\% of the general world facts per person. For comparison, AlphaEdit—an established parameter-efficient editing baseline—retained 75.21\% under similar conditions (98.78 / 99.83). Thus, SPIKE maintains general knowledge at a level comparable to editing models explicitly designed for knowledge preservation.
>
>
> Beyond general-knowledge retention, another widely discussed dimension of “generalization’’ in knowledge-editing research concerns robustness to paraphrased or rephrased queries, as exemplified in benchmarks such as zsRE. To assess this aspect, we also examined paraphrase-level generalization. Importantly, paraphrase robustness depends strongly on the intrinsic linguistic capacity of the base model, especially because personal facts typically fall outside the model’s pretraining distribution. As a result, the model must rely on its general semantic generalization abilities even before any personalization or editing method is applied. To illustrate this, we performed a small control experiment without using SPIKE: after training both GPT-J and LLaMA with the same initial personal fact (“Elle Quinn has a job of personal trainer”), we queried the base models using only the name “Elle Quinn.’’ A model with strong inherent generalization should recover a personal trainer under such paraphrased prompting. LLaMA3-8B-Instruct successfully did so, whereas GPT-J-6B failed. This indicates that paraphrase-level generalization is largely determined by the underlying LLM architecture rather than by SPIKE.
>
> Motivated by the reviewer’s separate question regarding LLaMA applicability, we then applied SPIKE to a LLaMA-7B model and evaluated paraphrased personal-fact queries (e.g., “Mike has a job of X’’ → “Mike works as X’’). SPIKE achieved 71.34\% paraphrase consistency on LLaMA, confirming that the method maintains updated personal facts across diverse formulations when built upon a model with sufficient linguistic capacity.
>
> We are currently extending additional metrics to all baselines and all datasets. Once these evaluations are completed, we will incorporate the results into the revised version of the paper.

---

> ### Author Response · Authors · 2025-11-21
> **Response to Weakness 7**
>
> > Weakness 7. The RAG baseline comparison in §5.3.1 is performed assuming 100\% retrieval success. This idealized setting does not reflect a true RAG system's performance; instead, it represents a "sufficient context" scenario.
>
> **Response to Weakness 7.**
>
> Thank you for this precise observation. We apologize for the confusion caused by our imprecise description of the context-provision setting. As you correctly noted, the experiment in Figure 3(a) was intended to illustrate an idealized scenario where retrieval is perfect. The result in Figure 3 (a) shows that a model given sufficient context fails to properly incorporate the updated facts. This empirical result motivates the need for representation-level activation intervention, as proposed in our method. We agree that the correct terminology for this setting is indeed “sufficient context.”

---

> > ### Comment · Reviewer_ykKB · 2025-11-27
> > **Response to Authors of Paper 18321**
> >
> > Thank you for adopting my review comments and providing detailed responses. I have raised my score to 6 and wish you all the best.

---

### Official Review · Reviewer_Mvpa · 2025-10-31

**Soundness:** 3
**Presentation:** 3
**Contribution:** 3
**Rating:** 6
**Confidence:** 4

**Summary:**

The paper tackles editing/updating user-specific facts in LLMs with minimal collateral changes by modeling personal info as a knowledge graph and specifying both where to intervene in the model and how to inject the update signal. It uses circuit analysis to show personal facts are localized in attention-head circuits and that targeting heads is far more effective than FFNs. They propose SPIKE aligns KG-triple deltas with internal representations and steers only the top-k important heads, improving accuracy while preserving locality without full-model fine-tuning.

**Strengths:**

1. Well-scoped problem + principled solution. The paper designs SPIKE to answer the two key questions—where to edit (personal-knowledge circuits) and how to inject updates (a delta derived from the KG)—with evidence that attention-head targets beat FFNs by a wide margin under similar parameter budgets.
2. SPIKE maintains high locality while integrating new facts and generalizes to unseen triples, outperforming a strong RAG baseline in that setting.
3. The paper probes which heads matter, how many heads to steer, and whether subgraph features help; removing subgraph features or replacing important heads degrades performance, and using too few heads hurts locality.

**Weaknesses:**

1.  External validity is limited by synthetic setups and assumptions. All experiments use synthetic personal-KG datasets (no real user data or user studies), and the approach assumes personal information is available as a well-formed Personalized Knowledge Graph to compute update deltas—conditions that may not hold in practice and raise open deployment questions despite the ethics discussion.
2. Missing a similar work as baseline. Knowledge Graph Tuning: Real-time Large Language Model Personalization based on Human Feedback.

**Questions:**

See weaknesses.

---

> ### Author Response · Authors · 2025-11-21
> **Response to Weakness 1**
>
> >Weakness 1. External validity is limited by synthetic setups and assumptions. All experiments use synthetic personal-KG datasets (no real user data or user studies), and the approach assumes personal information is available as a well-formed Personalized Knowledge Graph to compute update deltas—conditions that may not hold in practice and raise open deployment questions despite the ethics discussion.
>
> **Response to Weakness 1.**
>
> We appreciate the reviewer for highlighting this important point concerning external validity and practical deployment. We fully agree with the reviewer that PeaCoK-Ex and PerInfoKG are synthetic datasets whose entity representations were deliberately cleaned and normalized. In our original construction, descriptive and free-form modifiers present in the raw PeaCoK data—such as in (Arlo Hill, has a job of, heavy duty equipment operator who work long and hard) or (aids in the completion of large projects, is a social routine or habit of, heavy duty equipment operator who work hard at my job)—were removed to obtain a standardized entity form (e.g., “heavy duty equipment operator”). As correctly pointed out, this cleaning process reduces the heterogeneity and noise typical of personal data encountered in practical deployment scenarios.
>
> | Dataset            | Method | Accuracy | Locality | Total Score |
> |--------------------|--------|----------|----------|---|
> | PeaCoK-Ex-Noisy    | FT  | 95.12    | 9.42    | 17.14|
> | PeaCoK-Ex-Noisy    | AlphaEdit  | 6.1    | 96.4    | 11.47 |
> | PeaCoK-Ex-Noisy    | SPIKE  | 98.17    | 93.47    | 95.76|
> | PeaCok-Ex | SPIKE | 100 | 99.3 | 99.65|
>
> To assess whether our method remains effective under more realistic and less idealized conditions, we constructed PeaCoK-Ex-Noisy, which preserves the raw, unprocessed entity strings exactly as they appear in the original dataset. In this more deployment-relevant setting, the model must learn to inject and update the entire noisy entity, not merely a canonical label, thereby reflecting the kind of messy personal information systems would encounter in practice. Under this noisy regime, SPIKE attains 98.17 accuracy, 93.47 locality, and a total score of 95.76, demonstrating that our method continues to operate reliably even when applied to unnormalized, noisy personal knowledge. In contrast, AlphaEdit showed clear limitations under this noisy setting, as editing over long and irregular sequences was unstable, while full-weight finetuning substantially distorted previously acquired knowledge and resulted in marked locality deterioration. These results suggest that SPIKE’s alignment and update mechanism maintain robustness beyond sanitized synthetic environments, addressing concerns about the approach’s applicability in real-world deployment contexts. The corresponding experimental details have been incorporated into Section~5.3 (Extended Experiments Results) of the revised manuscript.

---

> > ### Author Response · Authors · 2025-11-21
> > **Response to Weakness 2**
> >
> > > Weakness 2. Missing a similar work as baseline. Knowledge Graph Tuning: Real-time Large Language Model Personalization based on Human Feedback.
> >
> > **Response to Weakness 2.**
> >
> > We thank the reviewer for pointing out Knowledge Graph Tuning (KGT) as similar work. We agree that KGT is a significant contribution, particularly in its approach to handling mutable personal knowledge.
> >
> > While both KGT and our work aim to adapt to evolving information, there is a clear distinction in the mechanism of knowledge injection:
> >
> > KGT (Symbolic & Contextual): KGT incorporates user feedback by modifying the symbolic structure of the Knowledge Graph (adding or removing triples). During inference, it relies on retrieving these updated triples and appending them to the input context. The model then performs in-context reasoning to incorporate this external information.
> >
> > Ours (Latent & Parametric): In contrast, our method operates directly on the model's internal state. We compute the vector difference between the representations of the updated and original knowledge, and inject this steering vector directly into the model's activations. This mathematically shifts the model's internal processing toward the new information, rather than relying on the model to attend to external context.
> >
> > Comparison with KGT: We acknowledge KGT as a valid baseline. Since KGT relies on retrieving information from the graph, its performance is naturally bounded by the accuracy of the retrieval step. To strictly evaluate the effectiveness of the knowledge injection mechanism itself (decoupled from retrieval variability), we compare against the performance ceiling of the retrieval-based approach. The ideal outcome of KGT is to retrieve the exact set of updated facts.
> >
> > This ideal state corresponds to our "Sufficient Context" baseline (Figure 3-a), where we manually provide the correct updated information as context. Therefore, our "Sufficient Context" baseline serves as a proxy for the idealized performance of KGT. The result in Figure 3 (a) shows that our method (0.73) outperforms this context-based upper bound (0.692), demonstrating that directly steering the model's internal representation is more effective than in-context learning, even under perfect retrieval conditions.
> >
> > We include KGT in our manuscript in the section discussing LLM personalization (Appendix A.8)

---

### Official Review · Reviewer_T68E · 2025-11-10

**Soundness:** 3
**Presentation:** 3
**Contribution:** 4
**Rating:** 8
**Confidence:** 4

**Summary:**

The paper introduces SPIKE (Steering for Personalized Knowledge Injection): a circuit-aware activation-steering method to update mutable, user-specific facts inside LLMs without full fine-tuning or pre-cached representation. The workflow is: (i) identify personal-knowledge circuits (a sparse network of attention heads) via head-importance scoring; ii) learn a knowledge-graph-LLM alignment module that maps a triple/subgraph change representing a factual change into a difference vector Δ. iii) add the Δ into the selected heads’ activations at run-time to realize the update. Experiments on two synthetic personal-KG datasets (PeaCoK-Ex, PerInfoKG) show high Accuracy (alignment with updated fact) and strong Locality (effect on unrelated/unchanged fact), outperforming editing, LoRA, circuit-selective finetuning on GPT-J and Qwen2.5-7B.

**Strengths:**

1.	Clearly identify the problem within existing solutions and address them with a novel approach: The paper identifies a clear, practical, and important gap in existing literature: finetuning approach is computationally expensive and risk catastrophic forgetting; knowledge editing like ROME and MEMIT requires access to the precached representation of original model’s knowledge, which is not accessible due to commercial and privacy concern. The paper proposes a clever approach by using an alignment module that only requires aligning a specific fact with its updating version during training, and use it to steer activation
2.	Conduct a thorough investigation on where to apply activation changes: The authors uses the HeadMap to investigate to identify a subset of attention head only, and ablation study was conducted to compare applying activation changes elsewhere; Another significant decision the authors clearly justified is the position to apply the alignment module (targeting attention heads over FFNs), and experimental evidence is provided. The experimental methodology is sound and well thought out.
3.	Generalization Capability: The experiment on "unseen triples (factual changes)" is a key strength. It shows that the KG-LLM Alignment Module learns a generalizable function for mapping knowledge changes to activation deltas, rather than just memorizing a fixed set of edits.

**Weaknesses:**

1.	Synthetic & Structured Data: The paper's primary weakness is its exclusive reliance on synthetic datasets - PeaCoK-Ex, PerInfoKG. These have clear relationship-level definitions of fact such as (Patient name, medical condition, disease) and simple representation of triples. However, the real-world factual representation is far more complicated than this and maybe ambiguous: for example, The doctor ran all the tests and said I'm perfectly healthy, but I still feel exhausted and ill all the time. In this example the facts are more difficult to extract and correspond to more complex relationships. This would require more triples or more complex construct to encode, potentially requiring a more complex alignment module.
2.	Implicit Assumption: A Global "Personal Circuit": The circuit discovery appears to identify a single, global "personal knowledge circuit" for each model, averaged over the entire dataset. This assumes that all personal facts (for all users, across all relations like 'Job', 'Medical Condition', 'Location') are encoded in the same sparse subgraph. This seems biologically and computationally implausible. A more likely scenario is that circuits are sparse but content-dependent or even user-dependent. The paper does not test this assumption.
3.	Limited scope on how the Locality is measured: The paper's experiments are focused on the triples contained within their two custom-built Personal Knowledge Graphs (PKGs). Locality is specifically defined as checking the model's accuracy on facts (triples) that were not part of the update. Hence, the Locality metric only measures forgetting of other personal facts, not general knowledge. This makes the approach seem more "lightweight" than it may be in practice.

**Questions:**

1.	On real world representation: The method is entirely dependent on clean, structured KG triples. What is your proposed path for applying SPIKE to real-world personalization, where user facts must be extracted from unstructured, messy, and often implicit text such as emails and chats?
2.	Activations of alignment module: the paper did not explicitly mention how the activation of alignment module is done. Is it similar to a RAG? Would the activation of alignment module be accurate if the an ambiguous question is asked, let’s say what is the daily routine of Mike, which likely needs the information of this occupation (pilot, in your illustrative example)?
3.	On training of the Alignment module: it seems that the training is one-off in your experiment. How would the training take place, suppose Mike’s occupation undergoes a third change (student->pilot->CEO)? Do you need to gather all the training data again and then train the module? Is there privacy concern in gathering all data from other people, since you suggest an alignment module on each user is prohibitively expensive, so all users need to share an same alignment module?

---

> ### Author Response · Authors · 2025-11-21
> **Response to Weakness 1 & Question 1**
>
> > Weakness 1. Synthetic \& Structured Data: The paper's primary weakness is its exclusive reliance on synthetic datasets - PeaCoK-Ex, PerInfoKG. These have clear relationship-level definitions of fact such as (Patient name, medical condition, disease) and simple representation of triples. However, the real-world factual representation is far more complicated than this and maybe ambiguous: for example, The doctor ran all the tests and said I'm perfectly healthy, but I still feel exhausted and ill all the time. In this example the facts are more difficult to extract and correspond to more complex relationships. This would require more triples or more complex construct to encode, potentially requiring a more complex alignment module.
>
> > Question 1. On real world representation: The method is entirely dependent on clean, structured KG triples. What is your proposed path for applying SPIKE to real-world personalization, where user facts must be extracted from unstructured, messy, and often implicit text such as emails and chats?
>
> **Response to Weakness 1 & Question 1**
>
> Thank you for this thoughtful and important point. We fully agree with the reviewers’ observation that our primary datasets, PeaCoK-Ex and PerInfoKG, are synthetic and based on clean, de-noised entity representations. As noted, this construction may underrepresent the complexity, ambiguity, and noise present in real-world personal knowledge.
>
> To directly address this concern, we conduct an additional experiment using a more realistic and noisy variant of the dataset. The original PeaCoK dataset contains many unstructured and descriptive entity forms. For example:
> (Arlo Hill, has a job of, heavy duty equipment operator who works long and hard), (aids in the completion of large projects, is a social routine or habit of, heavy duty equipment operator who works hard at my job)
>
> In PeaCoK-Ex, we previously removed such free-form descriptive suffixes (e.g., “who work long and hard”) and retained only the canonical entity (e.g., “heavy duty equipment operator”) to construct a structured, clean KG. As the reviewers correctly point out, this denoising process reduces dataset complexity.
>
> To examine whether SPIKE remains effective under more realistic noise conditions, we create a new dataset, PeaCoK-Ex-Noisy, where we retain the original noisy descriptive entities without any cleaning. Under this setting, a model must correctly incorporate and update the noisy string (e.g., “heavy duty equipment operator who works long and hard”), not just the canonical job name.
>
> | Dataset            | Method | Accuracy | Locality | Total Score |
> |--------------------|--------|----------|----------|---|
> | PeaCoK-Ex-Noisy    | FT  | 95.12    | 9.42    | 17.14|
> | PeaCoK-Ex-Noisy    | AlphaEdit  | 6.1    | 96.4    | 11.47 |
> | PeaCoK-Ex-Noisy    | SPIKE  | 98.17    | 93.47    | 95.76|
> | PeaCok-Ex | SPIKE | 100 | 99.3 | 99.65|
>
>
> On PeaCoK-Ex-Noisy, SPIKE achieves 98.17 accuracy, 93.47 locality, and a total score of 95.76, showing only a modest degradation relative to the clean PeaCoK-Ex results (99.65). These results indicate that SPIKE remains robust even when applied to noisy, unstructured, and more realistic KG representations, mitigating the reviewers’ concern regarding the synthetic nature of our datasets. In contrast, AlphaEdit struggled in this setting, as injecting noisy and long sequences proved unstable, and full-weight finetuning severely disrupted pre-existing knowledge, leading to substantial degradation in locality.
> The corresponding experimental details have been incorporated into Section 5.3 (Extended Experiments Results) of the revised manuscript.

---

> ### Author Response · Authors · 2025-11-21
> **Response to Weakness 2.**
>
> > Weakness 2. Implicit Assumption: A Global "Personal Circuit": The circuit discovery appears to identify a single, global "personal knowledge circuit" for each model, averaged over the entire dataset. This assumes that all personal facts (for all users, across all relations like 'Job', 'Medical Condition', 'Location') are encoded in the same sparse subgraph. This seems biologically and computationally implausible. A more likely scenario is that circuits are sparse but content-dependent or even user-dependent. The paper does not test this assumption.
>
>
> **Response to Weakness 2.**
> Thank you for raising this thoughtful point regarding the assumption of a single “global” personal circuit. We fully agree that user-dependent or relation-dependent circuits may exist, and such heterogeneity is entirely plausible from both computational and cognitive perspectives.
>
> We compare “global” and “personal” circuits in terms of effectiveness (by comparing accuracy and locality) and scalability. First, we compared the accuracy and locality in the PeaCoK-Ex dataset when applying “global” or “personal” circuits. The table shows the overall result. As shown, the performance difference between the two settings is negligible, indicating that both approaches are similarly effective in terms of accuracy and locality.
>
>
> | Circuit            | Accuracy | Locality |
> |--------------------|--------|----------|
> | Global    | 100  | 99.3    |
> | Personal | 100 | 99.4 |
>
> The distinction becomes more pronounced when considering scalability. “Personal” circuits require computing a new circuit for each incoming user before SPIKE can be applied, which incurs ongoing per-user overhead. In contrast, the global circuit can be maintained and reused as new users arrive. Its robustness to unseen users is further supported by the results demonstrated in Figure 3(a), where the global circuit remains effective even under an unseen test scenario.
>
> In summary, while both circuit types exhibit comparable effectiveness, the global circuit offers substantially better scalability. For this reason, we adopt the global circuit in our method. This content has been added to Section 4.1 and Appendix A.6 of the revised manuscript.

---

> ### Author Response · Authors · 2025-11-21
> **Response to Weakness 3.**
>
> > Weakness 3. Limited scope on how the Locality is measured: The paper's experiments are focused on the triples contained within their two custom-built Personal Knowledge Graphs (PKGs). Locality is specifically defined as checking the model's accuracy on facts (triples) that were not part of the update. Hence, the Locality metric only measures forgetting of other personal facts, not general knowledge. This makes the approach seem more "lightweight" than it may be in practice.
>
>
> **Response to Weakness 3.**
> We appreciate the reviewer for highlighting this important point. We fully agree that improving personalization at the cost of degrading the model’s general world knowledge would be undesirable and counterproductive. To directly evaluate this concern, we conducted an additional experiment measuring general-knowledge retention after applying SPIKE.
>
> Following prior work such as ROME, we used the''known-1000'' fact set—a collection of factual queries of general knowledge that GPT-2 is known to answer correctly (e.g., ''Catalonia belongs to the continent of''). Among these, we first identified a subset of 200 facts that our base model (GPT-J finetuned with only initial personal knowledge, init-LLM) answered correctly. We then applied SPIKE to inject the updated personal knowledge and re-evaluated the model on the same 200 general-knowledge questions.
>
> | Method     | Accuracy | Locality | Retention Rate |
> |------------|----------|----------|----------------|
> | SPIKE      | 100     | 100     | 73.95            |
> | AlphaEdit  | 98.78   | 99.83   | 75.21            |
>
>
> To evaluate the retention rate, we ran the entire training–evaluation workflow again from the original t1-LLM initialization. When SPIKE achieved 100\% accuracy and 100\% locality on the PKG update, it retained an average of 73.95\% of the general world facts per person. For comparison, AlphaEdit—another state-of-the-art parameter-efficient editing method—retained 75.21\% under a 98.78\% of accuracy and 99.83\%. Thus, SPIKE maintains general knowledge at a level comparable to editing models explicitly designed for knowledge preservation.
>
> These results indicate that the effects of SPIKE on general knowledge are not limited in practice and are comparable to those of existing lightweight editing approaches, suggesting that the method does not introduce disproportionate forgetting.
>
> We are currently extending this experiment to all baselines and all datasets. Once these evaluations are completed, we will incorporate the results into the revised version of the paper.

---

> ### Author Response · Authors · 2025-11-21
> **Answer to Question 2 & Question 3.**
>
> > Question 2. Activations of alignment module: the paper did not explicitly mention how the activation of alignment module is done. Is it similar to a RAG? Would the activation of alignment module be accurate if the an ambiguous question is asked, let’s say what is the daily routine of Mike, which likely needs the information of this occupation (pilot, in your illustrative example)?
>
> **Answer to Question 2.**
> We thank the reviewer for suggesting an interesting question about the effectiveness of the align module under an ambiguous question. In order to set an ambiguous question that likely needs the information of the occupation, we validated the efficacy of the align model to ask multi-hop questions such as (subject name, has a job whose characteristic is, ) or (subject name, has a job that requires, ). These questions need information about the occupation, but the answer is ambiguous. In several case studies, we found that by adding our steering vector from the align module, the LLM generates tokens related to the updated occupation. We include these case studies in the revised paper at Appendix A.9, discussing the capabilities of the align module in detail.
>
> > Question 3. On training of the Alignment module: it seems that the training is one-off in your experiment. How would the training take place, suppose Mike’s occupation undergoes a third change (student->pilot->CEO)? Do you need to gather all the training data again and then train the module? Is there privacy concern in gathering all data from other people, since you suggest an alignment module on each user is prohibitively expensive, so all users need to share an same alignment module?
>
>
> **Answer to Question 3.**
>
> >(a) How updates are handled when a user undergoes multiple changes (e.g., student → pilot → CEO)
>
> We agree that repeatedly retraining the alignment module for every new update would be impractical. This is precisely why we conduct the “unseen test” experiment in Figure 3-a of the manuscript, which directly simulates this scenario.
>
> In this experiment, the alignment module is trained once on a sufficiently large set of update examples. After this one-off training, we evaluate whether it can handle new, unseen updates without any further retraining. These unseen updates conceptually correspond to cases such as student → pilot → CEO: each new state transition is simply another update pair that was not explicitly seen during training.
>
> The results show that a single alignment module generalizes to these unseen updates, achieving 73\% accuracy, which is higher than the best in-context alternative (Sufficient Context : 69.2\%). This demonstrates that the alignment module does not need to be retrained for every new user update; it functions as a learned update rule that can be applied repeatedly.
>
>
> >(b) Whether training the alignment module requires aggregating all users’ personal data, and the associated privacy concerns
>
> We appreciate the opportunity to clarify this point. The sentence “Although one could assume a separate personalized LLM per individual … this is computationally prohibitive …” may have unintentionally created confusion about how the training data is handled.
>
> This statement refers only to the experimental setup, not to the deployment scenario.
> In our experiments, we used a single LLM instance due to memory constraints, and applied the alignment module separately for each individual. This was solely for experimental convenience.
>
> Importantly:
>
> 	- The alignment module does not store or aggregate raw personal information from multiple users.
> 	- It learns a general update function from representation-level differences (before vs. after), not from identifiable personal data.
> 	- Applying SPIKE does not require collecting or centralizing users’ personal KGs.
> 	- In deployment, each user’s update would be processed locally at the representation level, without exposing or pooling their underlying personal data.
>
> Thus, the method does not present additional privacy risks beyond those inherent in standard representation-based model adaptation.

---

### Author Response · Authors · 2025-12-03
**Summary of Author Responses**

We are grateful for the thoughtful and constructive reviews provided by all four reviewers. Their assessments highlight several strengths of our work, such as:

    (1) Novel Problem Framing and Conceptual Contribution
    (2) Methodological Soundness
    (3) Comprehensive and Well-Scoped Experimental Analysis
    (4) Strong Empirical Performance

At the same time, they raised important concerns regarding:

    (1) The exclusive use of synthetic data and concern about real-world applicability
    (2) Assumptions about circuit structure (why assigning a global circuit over a user-specific circuit)
    (3) The limited baselines and evaluations

> ### **Addressing (1): Real-world applicability of synthetic data.**

Three reviewers (T68E, Mvpa, ykKB) raised concerns about potential limitations stemming from the well-refined synthetic nature of our dataset. To directly address this, we regenerated the dataset without the noise-removal procedures used in the original construction. This produced a **realistic, noisy variant with unstructured and descriptive entity strings, reflecting the challenges of real-world personal information**. Using this noisy dataset, we demonstrated that SPIKE remains robust and effective even under realistic noise, substantially alleviating this concern.

> ### **Addressing (2): Questions about circuit structure.**

Reviewers also asked whether personal-knowledge circuits are user-specific or globally shared. We conducted additional analyses. In datasets with limited diversity across personal fields, the difference between user-specific and global circuits was negligible; in more diverse datasets, the global circuit variant performed slightly better, clarifying the reviewers’ questions.

> ### **Addressing (3): Expanding baseline and evaluation coverage.**

To broaden baseline diversity, we incorporated two additional editing baselines and explained why these methods struggle in our setting. We further validated SPIKE on a recent model (Llama-3-8B-Instruct), showing its applicability to modern architectures. To assess whether personalized updates overshadow general world knowledge, we introduced a new general-knowledge retention metric and showed that SPIKE maintains general knowledge comparably to editing methods explicitly designed for preservation.

&nbsp;

-------------

Through these discussions, we have strengthened the manuscript and directly resolved the main concerns raised by the reviewers. We appreciate that Reviewer ykKB acknowledged these improvements by raising their score by two points (to 6). Reviewer CZxM also posed follow-up questions that enabled deeper exploration, particularly regarding potential security risks associated with activation steering. Although this scenario is somewhat orthogonal to our problem setting, we proposed a feasible extension of SPIKE and supported it with empirical evidence.

&nbsp;

-------------

In summary, the reviewers collectively recognized the novelty, sound methodological foundation, rigorous experimentation, and strong performance of our approach, while also offering actionable concerns. We addressed these concerns thoroughly with additional experiments and analyses, and we are grateful that this led to improved assessments, most notably reviewer ykKB’s score increase from 4 to 6.

---

> ### Author Response · Authors · 2025-12-03
> **Additional Remarks**
>
> We have incorporated the retention rate for general knowledge as an additional metric in Table 2, along with an analysis of the corresponding experimental results. Furthermore, the experimental results regarding PeaCok-Ex-Noisy are included in Section A.7. Please note that the initial revisions (November 20) are marked in red color, while subsequent additional updates are highlighted in blue color.

---

### Meta-Review · Area_Chair_KKB1 · 2026-01-12

**Summary:**

All reviewers find the paper studying a timely and interesting problem. Their main concerns were about studying synthetic setting and limitation to factual updates. Authors included additional experimental results in their response. While the authors introduced a noisy variant of the dataset in the response to simulate unstructured text, the core concern remains. Real-world personal knowledge is often implicit, highly unstructured, and ambiguous (as noted by Reviewer T68E). It is not clear if the observations and methods in the paper extend to real wold setting. Hence I suggest rejection.

**Reviewer Concerns:**

Main reviewer concerns were around the limitation of the study to synthetic setting and limitation to factual updates.

**Reviewer Scores:**

ykKB 4-> 6
CZxM 4-> 4

---

### Decision · Program_Chairs · 2026-01-26

Reject